# The developing mouse coronal suture at single-cell resolution

D'Juan T. Farmer[1], Hana Mlcochova[2], Yan Zhou[2], Nils Koelling[2], Guanlin Wang[3,4], Neil Ashley[5], Helena Bugacov[1], Hung-Jhen Chen[1], Riana Parvez[1], Kuo-Chang Tseng [1], Amy E. Merrill[6], Robert E. Maxson Jr.[7], Andrew O. M. Wilkie [2], J. Gage Crump [1✉] & Stephen R. F. Twigg [2✉]

Sutures separate the flat bones of the skull and enable coordinated growth of the brain and overlying cranium. The coronal suture is most commonly fused in monogenic craniosynostosis, yet the unique aspects of its development remain incompletely understood. To uncover the cellular diversity within the murine embryonic coronal suture, we generated single-cell transcriptomes and performed extensive expression validation. We find distinct preosteoblast signatures between the bone fronts and periosteum, a ligament-like population above the suture that persists into adulthood, and a chondrogenic-like population in the dura mater underlying the suture. Lineage tracing reveals an embryonic *Six2+* osteoprogenitor population that contributes to the postnatal suture mesenchyme, with these progenitors being preferentially affected in a *Twist1+/−; Tcf12+/−* mouse model of Saethre-Chotzen Syndrome. This single-cell atlas provides a resource for understanding the development of the coronal suture and the mechanisms for its loss in craniosynostosis.

[1] Department of Stem Cell Biology and Regenerative Medicine, Keck School of Medicine, University of Southern California, Los Angeles, USA. [2] Clinical Genetics Group, MRC Weatherall Institute of Molecular Medicine, University of Oxford, John Radcliffe Hospital, Oxford, UK. [3] MRC Molecular Haematology Unit, MRC Weatherall Institute of Molecular Medicine, University of Oxford, John Radcliffe Hospital, Oxford, UK. [4] MRC WIMM Centre for Computational Biology, MRC Weatherall Institute of Molecular Medicine, University of Oxford, John Radcliffe Hospital, Oxford, UK. [5] Single cell facility, MRC Weatherall Institute of Molecular Medicine, University of Oxford, John Radcliffe Hospital, Oxford, UK. [6] Center for Craniofacial Molecular Biology, Ostrow School of Dentistry, University of Southern California, Los Angeles, USA. [7] Department of Biochemistry, Keck School of Medicine, University of Southern California, Los Angeles, USA. ✉email: gcrump@usc.edu; stephen.twigg@imm.ox.ac.uk

Cranial sutures are fibrous joints between the flat bones of the skull that act as zones of bone growth and absorbers of physical forces[1]. They comprise the leading edges of abutting cranial bones separated by mesenchymal tissue. New bone forms by intramembranous ossification in response to the expansion of the underlying brain[1–3]. Growth of the bony skull requires the proliferation and osteogenic differentiation of progenitor cells, as well as the maintenance of sufficient undifferentiated cells in the suture to ensure continued bone growth during fetal and postnatal stages. Environmental and/or genetic insults that disrupt the delicate balance of proliferation and differentiation result in premature fusion of cranial sutures, a condition known as craniosynostosis[4,5].

The coronal suture, which separates the frontal and parietal bones, is the suture most commonly affected in monogenic craniosynostosis[6]. The mouse has proven an effective model for the study of coronal synostosis[7–10]. During cranial development, the coronal suture and closely associated tissues are derived from three distinct populations: the supraorbital mesenchyme which will give rise to the cranial bones and suture mesenchyme, the meningeal mesenchyme, and non-osteogenic early migrating mesenchyme[11]. The meninges form between the brain and cranium and are essential for the development of both[12]. Coronal suture mesenchyme, derived largely from the mesoderm, forms a boundary between the neural crest-derived frontal and mesoderm-derived parietal bones[13,14]. Embryonic suture mesenchyme originates from *Gli1*-expressing cells that migrate away from the paraxial cephalic mesoderm at embryonic day (E) 7.5[15] and expand apically to sit between the lateral dermal mesenchyme and medial meningeal layers from E12.5 onwards[11]. Whereas these macroscopic developmental steps are well established, the cellular composition of the developing coronal suture remains poorly understood. Markers that label skeletal stem cells within postnatal sutures have been identified[16–19], yet none of these markers identify a distinct skeletal progenitor cell population at embryonic stages. Given increasing data for an embryonic etiology of craniosynostosis[20], identifying cell-type diversity in early forming sutures will be critical for understanding how embryonic progenitor dysfunction contributes to this birth defect. A better understanding of non-osteogenic populations will also inform how the meninges and ectocranial layers further contribute to suture patency[7,21,22].

To build a cell atlas of the embryonic coronal suture, we combined single-cell transcriptomics with highly resolved in situ analysis to catalog the cell types present at E15.5 and E17.5. In the ectocranial compartment, we uncovered multiple layers of distinct cell types, including a ligament-like population connecting the lateral aspects of the frontal and parietal bones that persists through adulthood. Within the multiple layers of the meninges, we revealed a dura mater population close to the bone with a chondrogenic signature, suggesting a latent capacity for chondrocyte differentiation. In the osteogenic population, pseudotime analysis revealed a putative *Erg*$^+$/*Pthlh*$^+$/*Six2*$^+$ progenitor that we found to be concentrated in the forming suture. These progenitors fed into two distinct preosteoblast trajectories, one concentrated at the growing bone tips and the other localized along the periosteum more distant from the suture. Lineage tracing showed the contribution of *Six2*+ cells to osteocytes within the bone (primarily frontal) and suture mesenchyme at both embryonic and postnatal stages. Expression analysis of genes associated with coronal synostosis highlighted the selective expression of many coronal synostosis genes, including *Twist1* and *Tcf12*, within osteogenic cells. Correspondingly, in *Twist1*+/−; *Tcf12*+/− mutants, a robust model for Saethre-Chotzen Syndrome coronal synostosis, *Erg*$^+$/*Six2*$^+$ progenitors were reduced as early as E14.5 yet ectocranial populations were relatively unaffected. This single-cell atlas reveals diversity within the osteogenic and non-osteogenic layers of the developing coronal suture, and

identifies some of the earliest markers for the embryonic osteoprogenitors affected in coronal synostosis.

## Results

**Mesenchymal heterogeneity captured by single-cell RNA sequencing.** To define the cellular composition of the embryonic coronal suture, we performed single-cell RNA sequencing of coronal sutures dissected from E15.5 and E17.5 mouse embryos. Dissections included small amounts of frontal and parietal bone, after removing the skin and brain (Fig. 1a). We filtered using Seurat 3 R-Package[23] and obtained 8279 cells at E15.5 (median of 2460 genes per cell) and 8682 cells at E17.5 (median of 3200 genes per cell) (Fig. 1b). We identified 14 cell clusters through unsupervised graph clustering of the two datasets combined. Osteogenic and mesenchymal cell types were identified based on the expression of broad mesenchyme/fibroblast (*Col1a1*) and osteoblast (*Sp7*) markers (Fig. 1b, dotted line; Supplementary Fig. 1a, b; Supplementary Data 1). The identities of clusters outside the osteogenic/mesenchymal subset were resolved using previously reported markers, and included chondrocytes, myeloid cells, mast cells, lymphocytes, pericytes, osteoclasts, endothelial cells, neurons, and glia (Fig. 1b, c). All the major cell types in our analysis were present at E15.5 and E17.5 (Supplementary Fig. 2a). Chondrocytes were especially abundant at E15.5 (Supplementary Fig. 2b), highlighting the close proximity of the E15.5 coronal suture to the chondrocranium. Myeloid cells were more abundant at E17.5, consistent with reports of increased myeloid differentiation during late embryonic stages[24] (Supplementary Fig. 2b).

To analyze the osteogenic and mesenchymal cell types that might comprise and support the coronal suture, we reclustered the osteogenic/mesenchymal population and obtained 14 clusters present at both E15.5 and E17.5 (Fig. 1d–f). Analysis of enriched genes for each cluster allowed us to assign probable identities to each cell type (Fig. 1g, Supplementary Data 2), which we validated by in situ hybridization as described below. We observed one ectocranial cluster strongly over-represented at E15.5 and a meningeal cluster over-represented at E17.5. Osteogenic cells were also more abundant in the E17.5 dataset, although it is unclear whether this reflects true biological differences versus differing cell capture between the dissections (Fig. 1a, f).

**Diversity of meningeal layers below the coronal suture.** The meninges are involved in the development of the cranium and underlying brain that they separate[12]. They comprise dura mater, arachnoid mater, and pia mater. To determine the identity of meningeal tissues included in our dissections, we utilized a recent transcriptomic study of murine E14 meninges as a guide[25]. Markers associated with the pia mater (*Ngfr*, *Lama1*, *Rdh10*) were not co-enriched in any of our clusters, consistent with the pia mater being removed with the brain during dissections (Fig. 2a). In contrast, markers enriched within the arachnoid mater (*Aldh1a2*, *Cldn11*, and *Tbx18*) were abundant in MG4, and dura mater markers (*Gja1*, *Fxyd5*, and *Crabp2*) in MG3 and MG4, and to a lesser extent MG1 and MG2 (Fig. 2a).

To resolve the identity of MG4, we performed in situ experiments for a highly specific MG4 marker, *Gjb6*, in combination with the arachnoid/dura mater marker, *Crabp2* (Fig. 2b). *Crabp2* and *Gjb6* overlapped below the bone with *Crabp2* single-positive cells (MG3) found above the *Crabp2*$^+$/*Gjb6*$^+$ domain (Fig. 2c). Immunofluorescence for Crabp2 and Gja1 at E17.5 confirmed that these arachnoid/dura mater markers are excluded from the pia mater (Supplementary Fig. 3a). *Rgs5*+ pericytes were interspersed with *Gjb6*+ cells in the MG4 arachnoid layer, consistent with the prominent vasculature extending from the border of the dura mater and through the arachnoid mater to the pia mater[26] (Supplementary Fig. 3c). In the

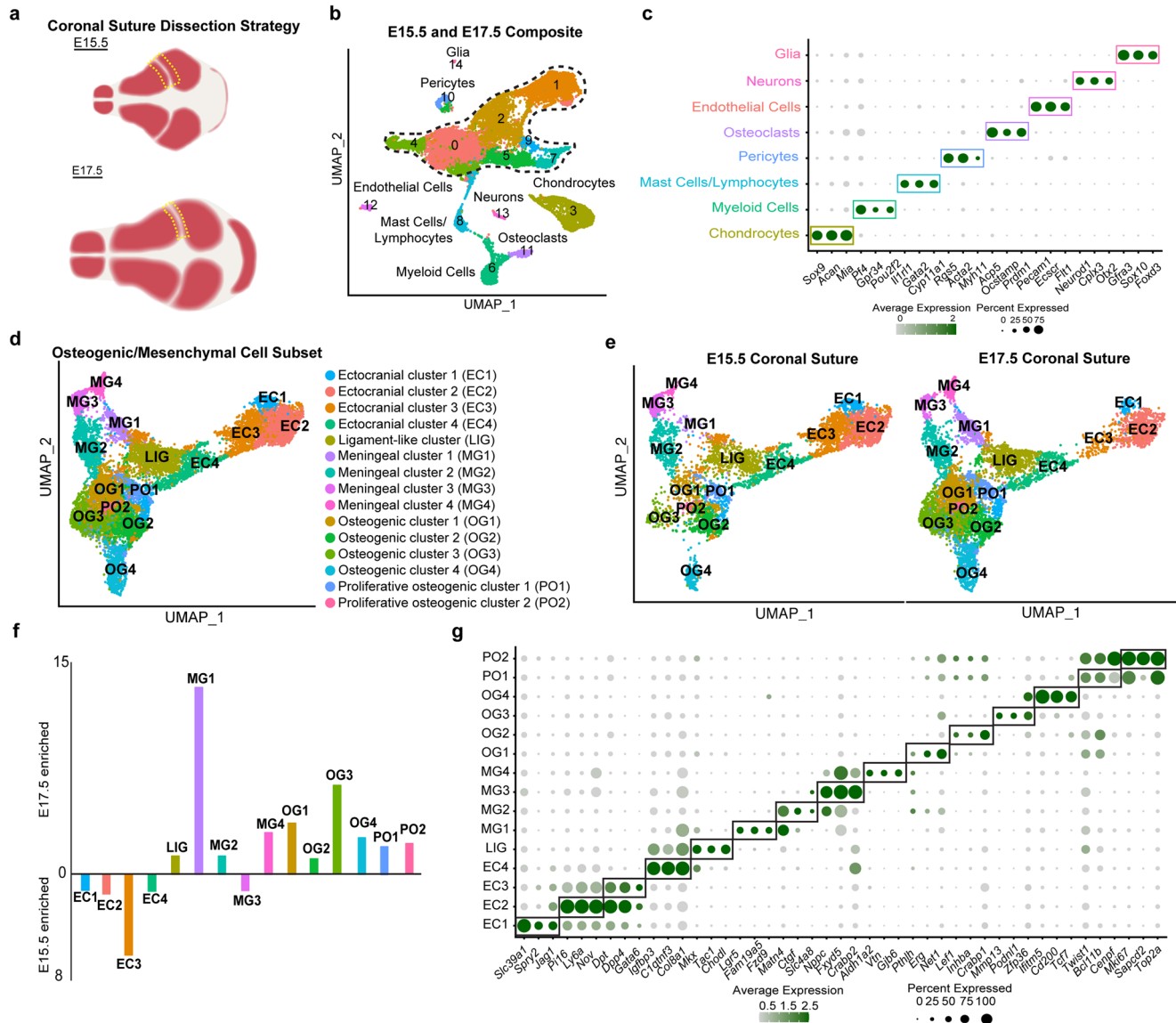

**Fig. 1 Single-cell RNA-sequencing analysis of the E15.5 and E17.5 coronal suture. a** Schematic of dissection strategy for E15.5 and E17.5 coronal sutures. Yellow dashed lines outline dissected regions. **b** Uniform Manifold Approximation and Projection (UMAP) plot of integrated E15.5 (8279 cells) and E17.5 (8682 cells) datasets. Osteogenic and mesenchymal cell types outlined by dashed lines. **c** Dot plot depicting selected markers (determined by adjusted *p*-value = 0 from the Wilcoxon Rank Sum test using FindAllMarkers in Seurat) enriched for each ancillary cell type outside of the osteogenic and mesenchymal population. **d** UMAP analysis of reclustered osteogenic and mesenchymal subset outlined in (b) resolves 14 clusters. **e** The osteogenic and mesenchymal subset separated by developmental stage. **f** Graphical depiction of the cluster proportions from the osteogenic/mesenchymal subset, plotted as ratio between E15.5 and E17.5 cells within each cluster. **g** Dot plot showing markers enriched for each cluster within the osteogenic/mesenchymal subset.

*Crabp2*+/*Gjb6*− layer (MG3), we also observed co-expression of *Crabp2* with *Nppc*, with a zone of *Nppc*+/*Crabp2*− cells (MG2) above (Fig. 2d). In addition, *Nppc* was co-expressed with the known dura mater marker *Fxyd5*[25] (Supplementary Fig. 3b, d). MG3 may represent the dural border cell layer that has been described by electron microscopy in adult meninges[27,28].

The *Nppc*+/*Crabp1*− population above *Nppc*+/*Crabp2*+ cells was also positive for *Matn4* and *Ctgf*, markers that only overlap with *Nppc* in MG2 (Fig. 2b, d, f, Supplementary Fig. 3b, e). *Matn4* expression does not overlap with the MG3/MG4 marker *Crabp2* (Fig. 2e), and chondrocytes in other sections express high levels of *Matn4* but not *Ctgf* (Supplementary Fig. 3f). We also detected some *Matn4* cells that were low or absent for *Nppc*, and these were situated closer to the suture and cranial bones than the *Nppc*-high cells (Fig. 2f). Our UMAP analysis indicates that these

*Matn4*+/*Nppc*− cells represent MG1 (Fig. 2b). MG1 expressed higher levels of the chondrocyte markers *Col2a1* and *Acan* than MG2, but not at the levels seen in bona fide chondrocytes (Supplementary Fig. 3g). These findings are consistent with MG1/MG2 representing a periosteal dura mater population[29] primed to form cartilage, with the MG1 population having a stronger chondrogenic-like program. These data reveal a diversity of mesenchyme cell types within the meninges, with those closest to the suture sharing features with chondrogenic cells and likely functioning as a specialized connective tissue bridging the cranial bones to the meninges (Fig. 2g).

**Diversity of ectocranial layers above the coronal suture.** Ectocranial mesenchyme guides the migration of early osteogenic cells

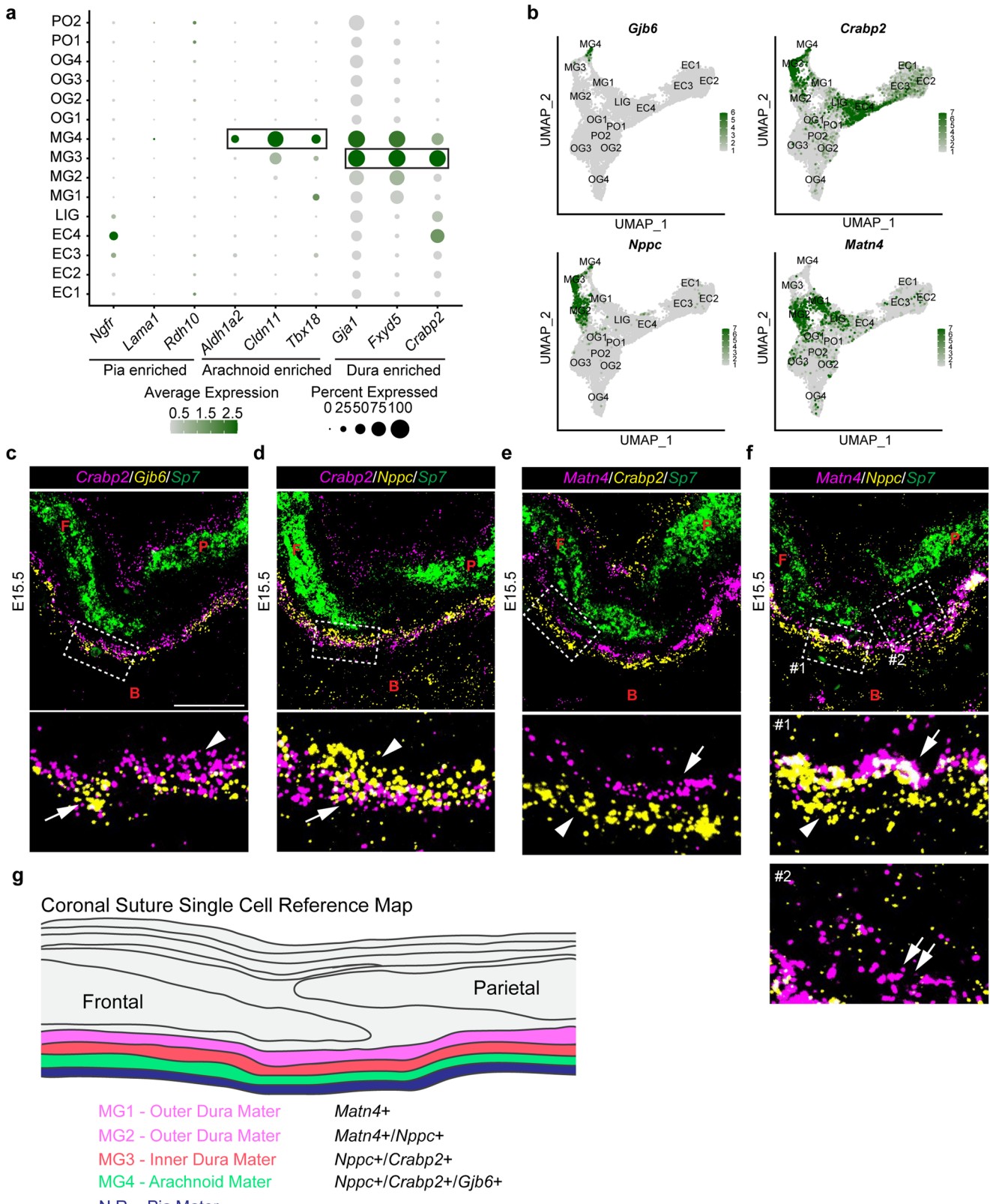

**Fig. 2 Diverse meningeal cell types are resolved by scRNA-seq. a** Dot plot depicting selected markers previously associated with the pia mater, the arachnoid, and the dura mater. **b** Feature plots of genes validated by in situ experiments. **c–f** Combinatorial in situ analysis of coronal sutures for indicated markers at E15.5. *Sp7* marks the frontal (F) and parietal (P) bones, except for insets (boxed regions) below. **c** *Crabp2* and *Gjb6*. Arrow, *Crabp2+/Gjb6+*; arrowhead, *Crabp2+*. **d** *Crabp2* and *Nppc*. Arrow, *Crabp2+/Nppc+*; arrowhead, *Nppc+*. **e** *Matn4* and *Crabp2*. Arrow, *Matn4+*; arrowhead, *Crabp2+*. **f** *Matn4* and *Nppc*. Arrow, *Matn4+/Nppc+*; arrowhead, *Nppc+*; double arrows, *Matn4+*. **g** Model summarizing gene expression patterns of meningeal layers captured from single-cell analysis. B, Brain. In situs were performed in biological triplicate. Scale bar = 50 μm; the upper panels in (**c–f**) are at the same magnification.

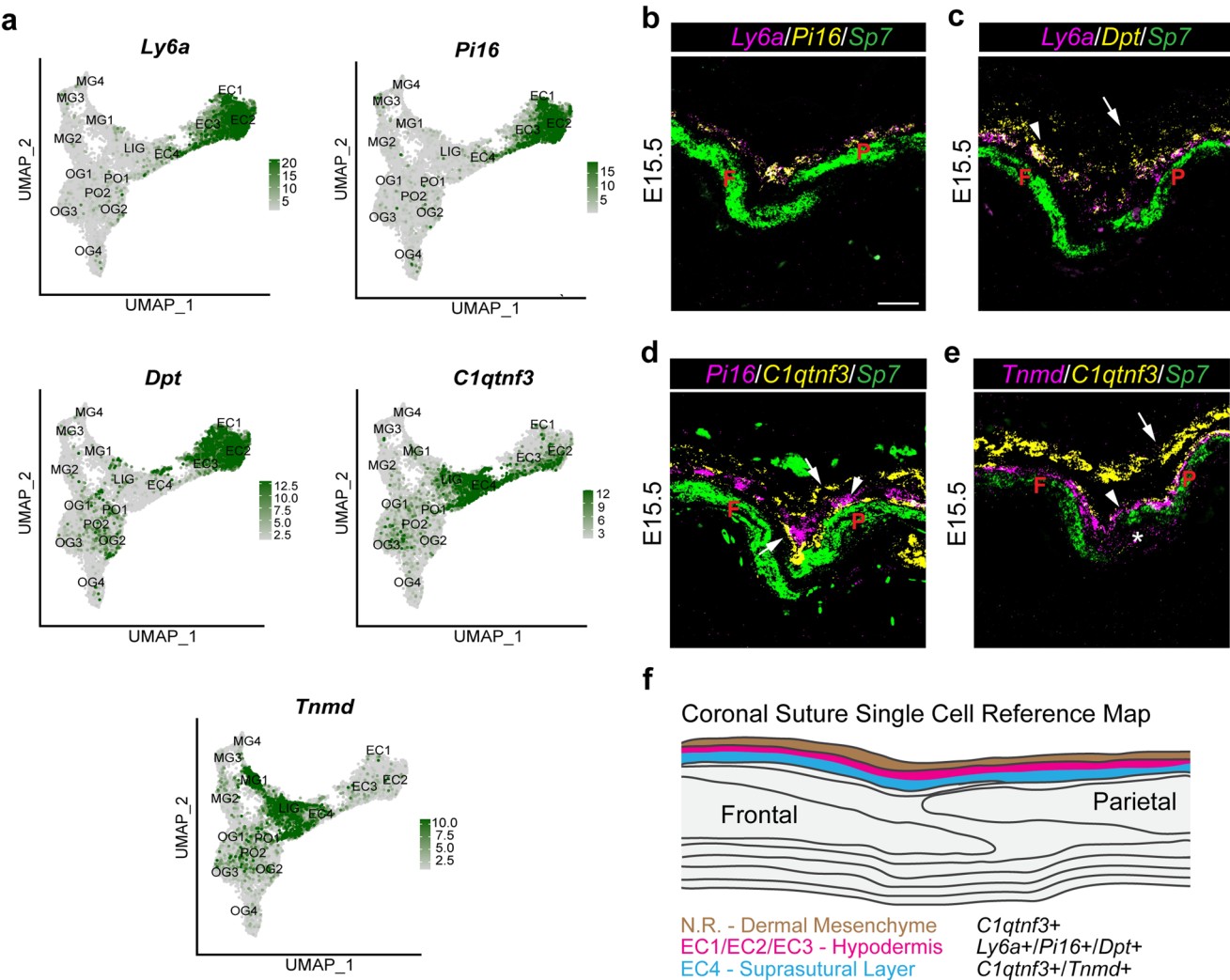

**Fig. 3 Multiple ectocranial layers overlay the coronal suture. a** Feature plots of genes validated by in situ experiments. **b**–**e** Combinatorial in situ analysis of coronal sutures for indicated markers at E15.5. *Sp7* marks the frontal (F) and parietal (P) bones. **b** *Ly6a* and *Pi16*. **c** *Ly6a* and *Dpt*. Arrow, *Dpt*+; arrowhead, *Ly6a*+/*Dpt*+. **d** *Pi16* and *C1qtnf3*. Arrows, *C1qtnf3*+ layers; Arrowhead, *Pi16*+ layer. **e** *Tnmd* and *C1qtnf3*. Arrow, *C1qtnf3*+; arrowhead, *Tnmd*+/*C1qtnf3*+. Asterisk, *Tnmd* expression in MG1. **f** Model summarizing gene expression patterns of ectocranial layers captured from single-cell analysis. In situs were performed in biological triplicate. Scale bars = 50 μm; images in (**b**–**e**) are at the same magnification.

and helps establish suture boundaries[7,21,22]. Consistent with analysis of the mouse frontal suture[30], we identified clusters enriched for markers of the dermis (*Ly6a*, *Dpt*; EC1-3) (Fig. 1g, Fig. 3a). *Ly6a* is expressed in hypodermis of neonatal mouse skin[31], and we observe co-expression of *Ly6a* with additional EC1-3 markers, *Pi16* and *Dpt*, within a single layer of mesenchyme above the skull bones (Fig. 3b, d; Supplementary Fig. 4a, b). Our previous work had identified expression of *Jag1* within both an ectocranial layer and suture mesenchyme cells[22], and scRNAseq analysis shows high expression of *Jag1* in EC1, and to a lesser extent EC2 and EC3 (Fig. 1g, Supplementary Fig. 4e). Co-expression of *Jag1* with *Pi16* confirms *Jag1* expression within the hypodermal layer (Supplementary Fig. 4f).

In addition to hypodermal EC1-3 layers, we also noted a prominent EC4 population defined by *C1qtnf3* and *Tnmd* expression in our UMAP analysis (Fig. 3a). In situ experiments revealed two *C1qtnf3*+ ectocranial layers on either side of the *Pi16*+ hypodermis (Fig. 3d, Supplementary Fig. 4c). However, only the *C1qtnf3*+ layer between the cranial bones and hypodermis expresses *Tnmd*, suggesting that this corresponds to EC4, which we term "suprasutural mesenchyme" (Fig. 3a, e, f; Supplementary Fig. 4d). It is likely that the outer *C1qtnf3*+ layer was removed along with the

skin during dissection and thus was not included in our single-cell analysis. Consistently, *Epha4*, which has also been shown to display ectocranial expression[7], was most strongly expressed in this outer *C1qtnf3*+ layer and not particularly enriched in clusters EC1-4 in UMAP analysis (Supplementary Fig. 4e, g). Thus, *Jag1* and *Epha4* appear to label distinct ectocranial layers.

**A ligament-like population above the coronal suture persists through adulthood.** In close association with the *C1qtnf3*+/*Tnmd*+ layer (EC4) and bridging the frontal and parietal bones across the top of the suture, we observed a *C1qtnf3*−/*Tnmd*+ population (LIG) enriched for genes associated with tendon/ligament development (e.g. *Scx*, *Tnmd*, *Mkx*, *Thbs2*, *Bgn*), as well as markers of smooth muscle (e.g. *Acta2*, *Tagln*, *Myl9*) (Supplementary Fig. 5a). Re-clustering of this population identified two transcriptionally distinct clusters: an *Mkx*+/*Acta2*− subset similar to tendons and ligaments and an *Mkx*+/*Acta2*+ subset more similar to the smooth muscle (Supplementary Fig. 5b, c). Two of the most specific markers for LIG were *Tac1*, a gene that encodes four neuropeptides including Substance P that is implicated in tendon mechanosensation[32], and *Chodl*, a membrane protein with expression in tendons of humans[33]

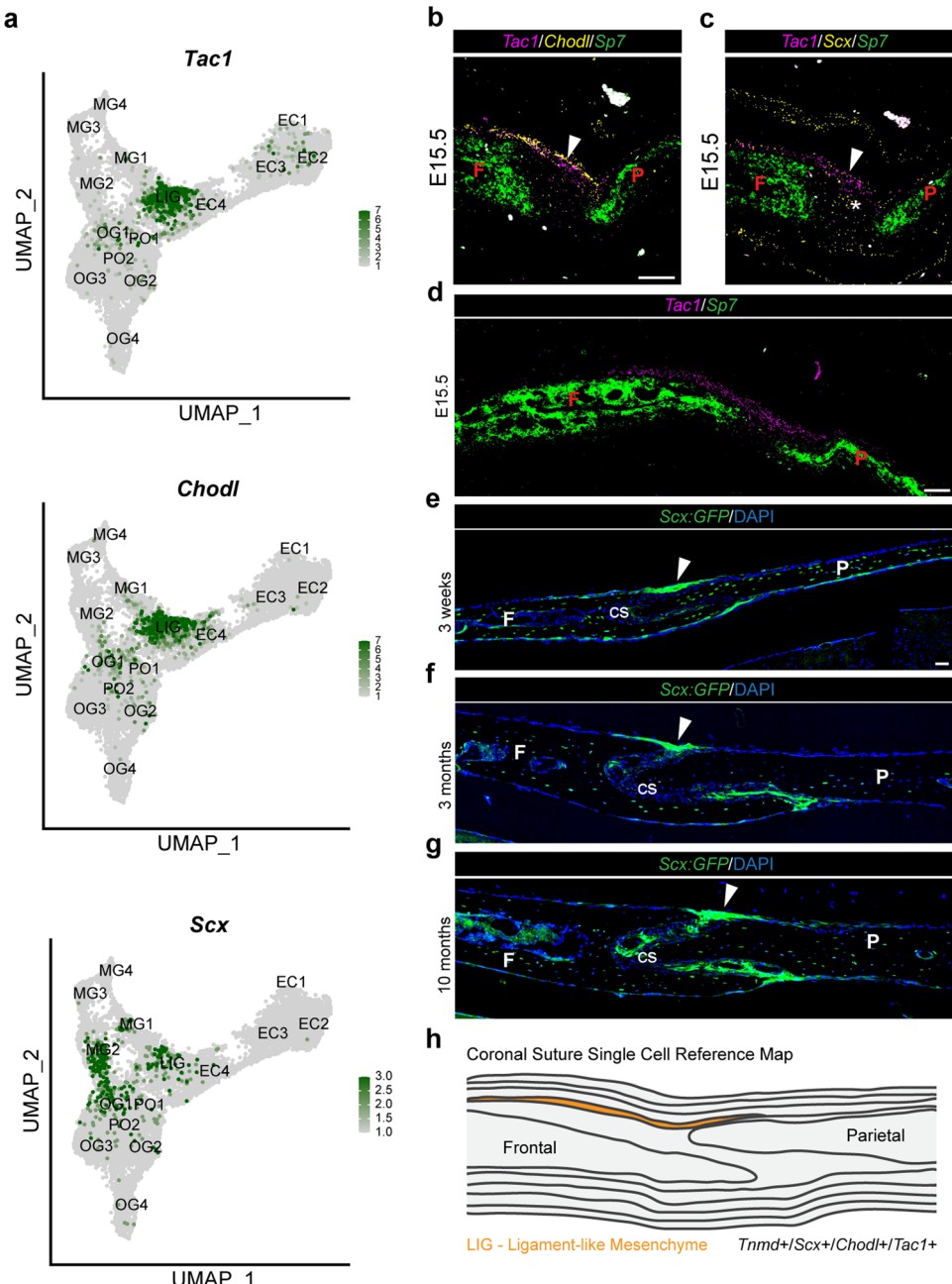

**Fig. 4 Ligament-like mesenchyme above the coronal suture. a** Feature plots of genes validated by in situ experiments. **b–d** Combinatorial in situ analysis of coronal sutures for indicated markers at E15.5. *Sp7* marks the frontal (F) and parietal (P) bones. Arrowheads denote the ligament-like population, and the asterisk suture mesenchyme weakly labeled by *Scx*. **e–g** At the indicated postnatal stages, *Scx-GFP* labels a ligament (arrowheads) in a similar position to the embryonic ligament-like population (*n* = 2 per stage). Nuclei are labeled by DAPI. **h** Model summarizing location of the LIG layer. cs = coronal suture. In situs were performed in biological triplicate. Scale bars = 50 μm; images (**b**, **c**) are at the same magnification, as are (**e–g**).

and mouse limbs[34] (Fig. 4a). *Tac1* expression is selectively enriched within the *Mkx*⁺/*Acta2*⁻ subset (Supplementary Fig. 5c). In situ hybridization for *Tac1* and *Chodl* revealed highly restricted expression in mesenchyme connecting the edges of the frontal and parietal bones above the coronal suture (Fig. 4b, d, Supplementary Fig. 5d). Although broader, in situ experiments showed co-expression of *Scx* and *Tnmd* within this layer (Fig. 4c; Supplementary Fig. 5e, f), and we observed only minimal overlap between *C1qtnf3* and *Tac1* expression (Supplementary Fig. 5g). Thus, the LIG cluster represents a transcriptionally distinct and spatially confined subset of the ectocranium overlying the coronal suture. At postnatal stages (3 weeks, 3 months, 10 months), analysis of the tendon/ligament

*Scx:GFP* reporter revealed long-term maintenance of this ligament-like structure (Fig. 4e–h).

**Distinct osteoblast trajectories in the suture versus periosteum.** The suture is a source of new osteoblasts for skull bone expansion. To better understand the trajectories of osteogenesis within and around the suture, we identified six osteogenic clusters (OG1, OG2, OG3, OG4, PO1, PO2) based on the expression of the osteoblast transcription factors *Runx2* and *Sp7*[35–37], as well as *Dlx5* and *Dlx6*[38–40], and *Ifitm5* and *Dmp1*[41,42] (Supplementary Fig. 6). Pseudotime analysis of these clusters using Monocle3

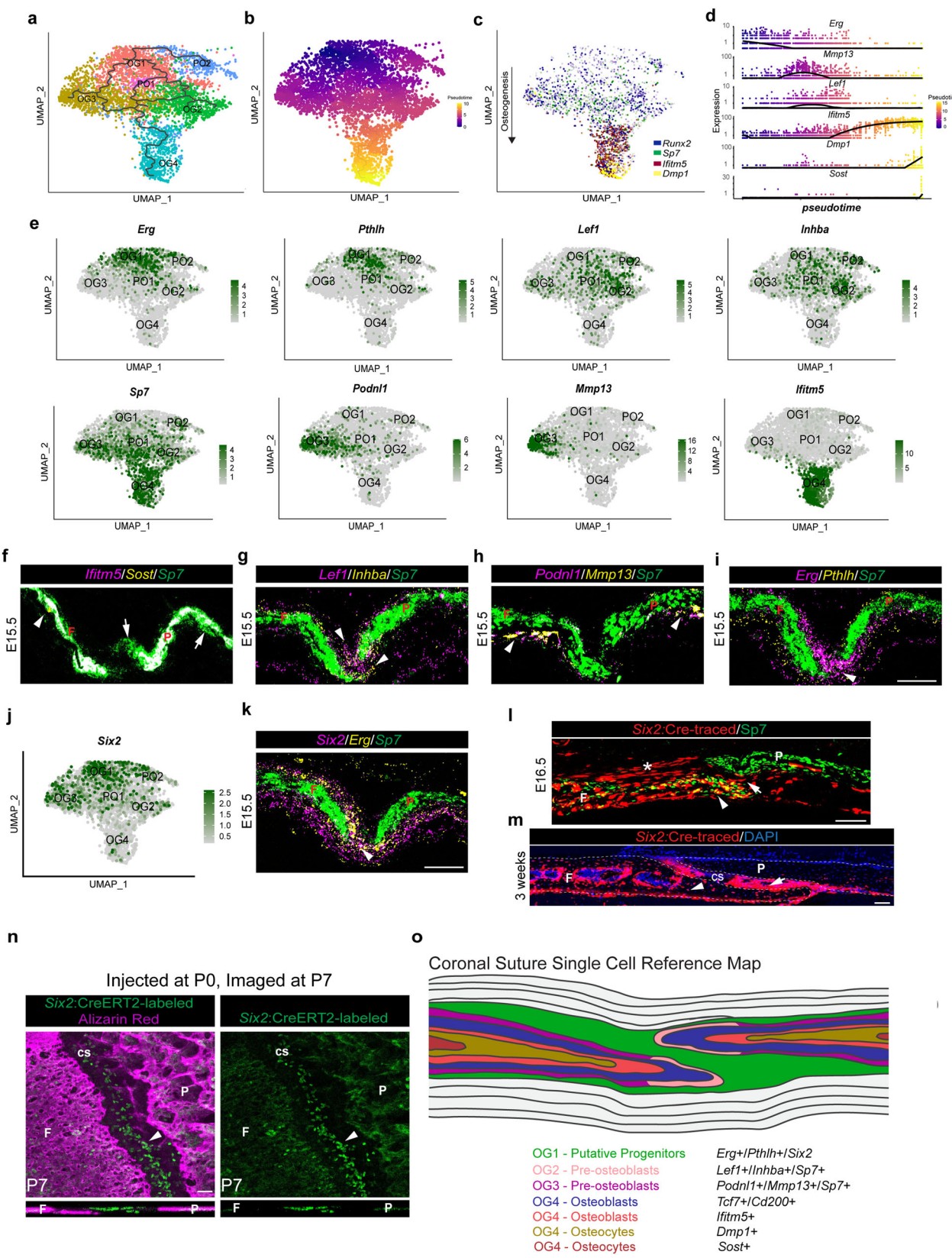

predicted OG1 to be the earliest lineage cells, followed by two proliferative branches (PO1, PO2) and distinct OG2 and OG3 trajectories that converged on the osteoblast cluster OG4 (Fig. 5a, b). These pseudotime trajectories were reflected by increased expression of *Runx2* and *Sp7* from OG1 to OG4 and accumulation of the mature osteoblast markers *Ifitm5* and *Dmp1* in OG4 (Fig. 5c). Prominent cluster markers included *Erg* and *Pthlh* (OG1), *Lef1* and *Inhba* (OG2), *Mmp13* and *Podnl1* (OG3), and *Ifitm5* and *Dmp1* (OG4) (Fig. 5c, e). Pseudotime visualization shows *Erg* (OG1) appearing earliest and shutting down as *Lef1*

**Fig. 5 scRNA-seq captures various subtypes of osteoblasts at the coronal suture. a** Lineage analysis and (**b**) pseudotime analysis of the osteoblast subset using Monocle 3. The gray lines in (**a**) indicate the lineage relationship between cells in the trajectory. In (**b**) cells further along the single-cell trajectory are indicated by progressively lighter colors. **c** Feature plot of selected markers within the osteogenic subset. **d** Expression plots of selected genes across pseudotime. **e, j** Feature plots of genes validated by in situ experiments. **f–i, k** Combinatorial in situ analysis of coronal sutures for indicated markers at E15.5. *Sp7* marks the frontal (F) and parietal (P) bones. **f** *Ifitm5* and *Sost*. Arrows indciate *Sp7+* expression in presumptive newly formed osteoblasts, and the arrowhead *Sost* expression. **g** *Lef1* and *Inhba*. Arrowheads mark *Lef1+/Inhba+* expression near bone tips. **h** *Podnl1* and *Mmp13*. Arrowheads mark *Podnl1+/Mmp13+* expression in periosteum distant from suture. **i** *Erg* and *Pthlh*. Arrowhead marks *Erg+/Pthlh+* expression in suture mesenchyme. **k** *Six2* and *Erg*. Arrowhead marks *Six2+/Erg+* expression in suture mesenchyme. **l, m** Immunostaining for Sp7 and tdTomato in a *Six2*:Cre;Ai9 coronal suture at E16.5 (*n* = 3) and DAPI staining of a Six2:Cre;Ai9 coronal suture at 3 weeks (*n* = 4). Asterisk, labeled cells above the frontal bone. Arrows, coronal suture. Arrowheads, osteocytes. Dashed lines outline bones. **n** Images of the coronal suture (cs, arrowhead) and adjacent bones from a P7 *Six2:CreERT2; R26-CAG-LSL-Sun1-sfGFP-myc* mouse treated with tamoxifen at P0 (*n* = 3). Converted cells display strong nuclear GFP; the diffuse green signal in the bones is autofluorescence. Top views are from above the cranium, with digital cross-sections through the suture shown below. **o** Model summarizing gene expression patterns within the developing bones and coronal suture mesenchyme. In situs were performed in biological triplicate. Scale bars = 50 μm; images (**f–i**) are at the same magnification.

(OG2) and *Mmp13* (OG3) become expressed, followed by their extinguishment and progressive appearance of *Ifitm5*, *Dmp1*, and *Sost* (OG4) (Fig. 5d).

To interrogate the spatial relationships between the osteogenic clusters, we first assessed markers of osteoblasts (OG4). The mature osteocyte marker *Sost*, which labels the oldest cells in pseudotime, was expressed in only a few osteocytes distant from the suture (Fig. 5f, Supplementary Fig. 7a, e). In addition, we observed co-expression of *Sp7* with *Ifitm5* throughout most of the bone, except for the leading tips and the surfaces of bones which are *Sp7*-positive only, consistent with bone growth from both the leading edges and the periosteal surfaces (Fig. 5f, Supplementary Fig. 7a). Markers for OG2 (*Lef1*, *Inhba*) and OG3 (*Podnl1*, *Mmp13*) displayed expression along the outer domains of the *Sp7+* bone surface, consistent with a pre-osteoblast identity. Whereas *Lef1+/Inhba+* cells were most abundant at the edges of the growing bones near the suture, *Podnl1+/Mmp13+* cells were enriched on bone surfaces further away from the suture (Fig. 5g, h; Supplementary Fig. 7b, c). Similarly, we observed protein expression of the osteoblast marker Cd200 (enriched in OG4) in bone and the Wnt-responsive transcription factor Tcf7 (enriched in OG2) along the bone tips and periosteal surfaces[43–45] (Fig. 1g, Supplementary Fig. 8a–c). OG2 may therefore represent specialized suture-resident pre-osteoblasts, and OG3 periosteal pre-osteoblasts more generally found on bone surfaces.

**Contribution of Six2+ osteoprogenitors to osteocytes and postnatal suture mesenchyme.** Pseudotime analysis places OG1 at the apex of the differentiation trajectory at the coronal suture (Fig. 5a–d). In situ validation showed co-expression of OG1 markers *Six2*, *Erg*, and *Pthlh* (Fig. 5e, j) within suture mesenchyme and extending along the edges of the frontal and parietal bone tips (Fig. 5i, k, Supplementary Fig. 7d). Interestingly, *Erg* was asymmetrically distributed along the bones, with stronger expression above the frontal and below the parietal bone. *Gli1* and *Prrx1* showed similar asymmetric expression above the frontal bone, and *Six2* and *Gli1* below the parietal bone (Fig. 5k, Supplementary Fig. 9a, b). These findings highlight the asymmetric distribution of the earliest osteogenic cells around the bone fronts, which may play a role in ensuring the later reproducible overlap of the parietal over the frontal bone.

Markers of the proliferative clusters PO1 and PO2 (*Mki67*, *Cenpf*, *Top2a*) overlapped with the OG1 marker *Erg* and OG2 markers *Lef1* and *Inhba* (Fig. 1g, Fig. 5e, Supplementary Fig. 7f). In situ validation revealed *Cenpf* expression around the tips of the growing bones and extending into the sutures at E15.5, although in comparison to *Erg* it appeared largely absent from the central part of the suture (Supplementary Fig. 7g). These data suggest that, distinct from OG1 osteoprogenitors, proliferative pre-osteoblasts are concentrated at the bone tips, consistent with previous evidence for proliferative osteogenic cells at the leading

edges of the cranial bones[20]. The spatial organization between putative progenitors, suture-resident pre-osteoblasts, and osteoblasts was confirmed with double in situ between markers for OG1 (*Pthlh*), OG2 (*Podnl1*), and OG4 (*Ifitm5*) (Fig. 5o, Supplementary Fig. 7h, i).

We next examined embryonic and postnatal labeling of suture cells with *Six2-Cre* and the Ai9 TdTomato reporter, as *Six2* is co-expressed with *Erg* in the coronal suture mesenchyme at E14.5 and E15.5. At E16.5, we observed labeling of mesenchyme in the coronal suture, which was asymmetrically distributed above the frontal and below the parietal bone, consistent with *Six2* expression (Fig. 5l). We also observed labeling of Sp7+ osteoblasts, primarily within the frontal bone. At 3 weeks after birth, we observed extensive contribution of *Six2-Cre*-labeled cells to coronal suture mesenchyme, frontal bone osteocytes, and parietal bone osteocytes close to the suture (Fig. 5m). Next, we used conditional *Six2*-CreERT2 mice, in combination with the R26-CAG-LSL-Sun1-sfGFP-myc reporter driving nuclear Sun1-GFP expression, to assess contributions of *Six2+* osteoprogenitors to postnatal coronal suture mesenchyme. Following treatment with tamoxifen at P0, we observed extensive contribution of labeled cells to coronal suture mesenchyme at P7, as well as scattered cells associated with the growing bone fronts that we interpret as osteoblasts (Fig. 5n). These findings are consistent with *Six2+* embryonic progenitors giving rise to postnatal suture mesenchyme.

**Overlapping cell types between the coronal and frontal suture.** In order to interrogate how differences in cell populations may underlie the differential sensitivity of the coronal and frontal sutures to craniosynostosis, we compared our dataset to a recently published dataset for the frontal suture[30]. Co-clustering of our E17.5 coronal suture dataset with the published E18.5 frontal suture dataset and UMAP analysis revealed 15 clusters, all of which had contributions from both datasets (Fig. 6a–c). For example, similar hypodermis ("EC1-3" in coronal versus "HD" in frontal), dura mater ("MG1-4" versus "DM"), and osteoblast ("OG4" versus "OB") populations were identified. The vast majority of marker genes for the coronal and frontal suture datasets also displayed similar expression in shared clusters between datasets (Fig. 6d, e). Of note, *Acta2+/Tagln+* cells formed a separate cluster (cluster 11) from *Tac1+/Chodl+* cells (cluster 5), consistent with re-clustering analysis of the LIG population (Supplementary Fig. 5a–c). Although not previously resolved in the frontal suture publication, we also note that the frontal suture dataset contains both OG3-like and OG2-like pre-osteoblasts.

Whereas we identify putative osteoprogenitors in both coronal and frontal suture datasets, the markers used to identify these appear to differ in their expression levels. In particular, we find

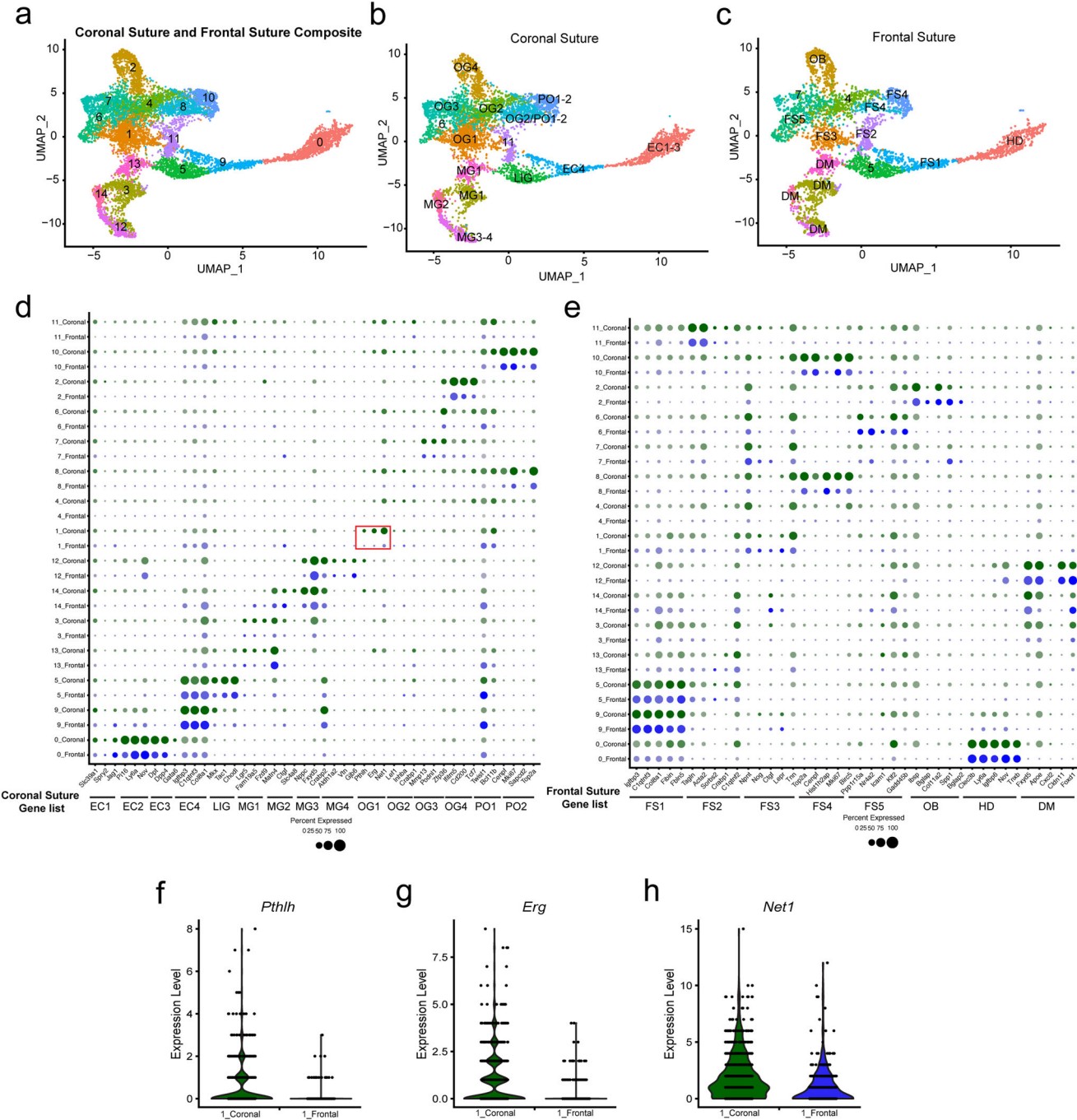

**Fig. 6 Comparison of frontal and coronal datasets. a** UMAP analysis of osteogenic and mesenchymal cells from integrated E17.5 coronal and E18.5 frontal datasets. **b**, **c** Cell clusters separated by suture type and labeled with respective names. **d**, **e** Dot plot showing the intersection of cluster markers from the coronal suture datasets (Fig. 1g) and for the frontal suture datasets from Holmes et al.[30], mapped onto to integrated dataset split by suture. Red box labels OG1 markers. **f–h** Violin plots highlight enriched expression of OG1 markers *Pthlh*, *Erg*, *Net1* in cluster 1 cells from the coronal versus frontal suture.

both higher representation and expression levels of *Erg* and *Pthlh*, and to a lesser extent *Net1*, in osteoprogenitors of the coronal versus frontal suture (Fig. 6d, f–h). There are also prominent differences in the spatial distribution of osteoprogenitors between sutures. In contrast to the asymmetric distribution of OG1 osteoprogenitors at the coronal suture, the FS3 osteoprogenitor marker *Npnt* was restricted to the medial aspects of the frontal suture and within the developing bone[30]. There are also differences in the spatial distributions of other cell types. The EC4 suprasutural population is distributed above the bones in the coronal suture, yet the comparable FS1 population sits within the

frontal suture. This comparison suggests that, while similar cell types are likely present at both types of sutures, gene expression and in particular spatial organization of cell types is suture-specific.

**Signaling interactions between mesenchymal layers captured from single-cell analysis.** To gain insights into potential signaling between osteogenic cells and adjacent ectocranial and meningeal mesenchyme, we surveyed our datasets for expression of ligands and receptors (Supplemental Fig. 10a, b). In OG4 osteoblasts, we observed enrichment of *Tgfb1*, *Bmp3*, *Bmp4*, *Ihh*, and *Pdgfa*, with

Ihh[46] and Pdgfa[47,48] known to be secreted by osteoblasts. In the ectocranial clusters EC4 and LIG, which are situated closest to the suture, we detected enrichment of *Tgfb3*, *Fgf9*, *Fgf18*, *Wnt5a*, *Wnt9a*, *Wnt11*, and *Igf1* ligands. In the meningeal clusters closest to the suture (MG1, MG2), we also detected *Tgfb3* and *Igf1* ligand expression, as well as *Tgfb2* that has previously been shown to be a critical meningeal-derived factor for suture morphogenesis[49]. Reciprocally, we detected expression of the Tgfβ receptor *Tgfbr3*, the Fgf receptors *Fgfr1* and *Fgfr2*, the Wnt receptors *Fzd1*, *Fzd2*, and *Fzd6*, the Ihh receptor *Ptch1*, and the Pdgfa receptor *Pdgfra* in various osteogenic clusters (Supplemental Fig. 10b). To capture signaling interactions in an unbiased approach, we interrogated our data using the CellPhoneDB package[50], scoring the ligand-receptor pairings between all clusters (Supplemental Fig. 10c). Interactions between osteogenic clusters were notably weak, highlighting the potential for interactions of osteogenic and non-osteogenic cell types in coronal suture regulation. Consistent with their spatial organization, the mesenchymal populations closest to the suture (EC4 above, MG1/MG2 below) had the strongest interactions with osteogenic cells. Of the ligand-receptor interactions that were identified between neighboring non-osteogenic and osteogenic clusters, several pathways are relevant to coronal suture development, including Fgf, Tgfβ, and Wnt signaling[51] (Supplementary Fig. 11).

**Enrichment of coronal synostosis gene expression in osteogenic cell clusters.** To determine if genes underlying craniosynostosis in humans[51–53] coalesce within a specific cell type, we mapped coronal synostosis genes onto our dataset. We noted that 9/16 coronal synostosis genes were enriched in several osteogenic clusters, in particular PO1 and PO2, suggesting misregulation of osteoblast differentiation and/or proliferation as a common mechanism of coronal synostosis. For example, two coronal synostosis genes *Cdc45* and *Esco2*, encoding DNA replication factors[54,55], were highly enriched in PO1, and *Fgfr1*, *Fgfr2*, *Tcf12*, *Twist1*, and *Zeb2* were enriched in both proliferative and osteogenic clusters, in particular osteoprogenitor cluster OG1 and suture-resident cluster OG2. To assess whether the coronal synostosis gene set had non-random distribution across suture cell types, we assessed the distribution of coronal gene set average expression, versus a randomized gene set, between grouped osteogenic, ectocranial, and meningeal clusters. Coronal synostosis genes were enriched in osteogenic clusters, and to a certain extent in meningeal clusters (Fig. 7b, Supplementary Fig. 12a). This analysis demonstrates a preferential enrichment of genes associated with coronal synostosis in osteogenic clusters.

**Selective loss of osteoprogenitors in a mouse model of Saethre–Chotzen Syndrome.** We had previously suggested that progenitor dysfunction plays a central role in the coronal synostosis observed in a *Twist1*[+/−]; *Tcf12*[+/−] mouse model for Saethre–Chotzen syndrome, although we had lacked clear markers to define embryonic osteoprogenitors in these studies[9,20]. The observed enrichment of *Twist1* and *Tcf12* in osteogenic clusters, including OG1, suggested that these transcription factors might regulate osteoprogenitor number. Further, in situ validation confirmed co-expression of *Twist1* with the proliferative marker *Cenpf* and the OG1 marker *Erg* in suture mesenchyme, with *Twist1* displaying the same asymmetric expression above the frontal bone as *Erg* (Fig. 7d, e, Supplementary Fig. 12b, c). In *Twist1*[+/−]; *Tcf12*[+/−] mice, we observed reduced expression of *Erg* at E14.5 and *Six2* at E14.5 and E15.5 in the coronal suture domain, yet ectocranial expression of *Pi16* and *Tac1* were unaffected (Fig. 7e–i). These findings point to a specific role of Twist1

and Tcf12 in the specification and/or maintenance of OG1 coronal suture progenitors as early as E14.5.

**Discussion**

Proper formation of the coronal suture is a complex process requiring the coordinated development of multiple tissue types, which is reflected in the diverse genetic heterogeneity underlying coronal synostosis. Understanding the mechanisms that cause pathogenic suture fusion in distinct syndromic forms of synostosis has been limited by our incomplete knowledge of the cellular diversity of individual types of sutures. In an effort to bridge this gap, we have generated a comprehensive spatial and transcriptomic map of the cell types that comprise and support the embryonic coronal suture.

The relationship of the embryonic progenitors that build the coronal suture with the adult sutural stem cells that grow the cranium remains debated. Recent work on postnatal sutures has demonstrated that the suture mesenchyme houses resident skeletal stem cells that produce osteoblasts to grow skull bones[16–18,56]. However, these markers (*Gli1*, *Prrx1*, *Axin2*) broadly label mesenchyme throughout the embryonic skull, making them unsuitable for identifying osteogenic cells at earlier stages. Here we identify *Erg*, *Six2*, and *Pthlh* as some of the earliest markers for the putative osteoblast progenitors concentrated in the embryonic coronal suture. Using *Six2-Cre* and *Six2*-CreERT2, we find that similar progenitors can be found at postnatal stages, where they contribute to osteocytes embedded in the neighboring bones. This supports a model in which embryonic *Erg*[+]/*Six2*[+]/*Pthlh*[+] progenitors contribute to postnatal sutural stem cells. Interestingly, *Pthlh* is also a marker for chondrocyte stem cells in the femoral growth plate[57], suggesting that Pthlh signaling could be a more general regulator of skeletal progenitors.

As opposed to the adult skull where stem cells are tightly restricted to suture mesenchyme, embryonic *Erg*[+]/*Six2*[+]/*Pthlh*[+] progenitors extend away from the suture along the surfaces of the frontal and parietal bone tips. Interestingly, these progenitors are asymmetrically distributed, with more cells along the lateral surface of the frontal bone and medial surface of the parietal bone. This asymmetric organization may ensure the reproducible architecture of the coronal suture, with the parietal bone consistently overlapping above the frontal bone. We show that asymmetric distribution of *Erg* and *Six2* is lost in *Twist1*[+/−];*Tcf12*[+/−] mutants, and in a companion study that similar asymmetric distribution of *Grem1*[+] progenitors is lost in *Tcf12*[−/−] mutants, consistent with bones meeting end-on-end and fusing in these craniosynostosis models[58]. These results highlight the power of our single-cell data to identify key embryonic cell types affected in models of craniosynostosis.

Our data also revealed distinct routes of osteoblast differentiation through either suture-resident pre-osteoblasts or periosteal cells more broadly distributed along the surfaces of bones. Suture-resident pre-osteoblasts are enriched for Wnt-related genes, perhaps reflecting the known involvement of genes from this pathway in coronal suture development[17,59], as well as proliferative markers such as *Cenpf*. These findings support a model in which proliferative bone growth from the coronal suture extends the lengths of bones, whereas more restrained osteoblast differentiation along the bony surfaces increases the thickness of the cranial bones, particularly at postnatal stages[60].

The underlying meninges are important in sutural and cranial development[12,61], and we uncovered multiple distinct layers associated with the coronal suture. In particular, we resolved the dura mater into a dural border cell layer and two layers consistent with periosteal dura[27,28]. Interestingly, the periosteal dura layer, which sits closest to the suture and cranial bones, was

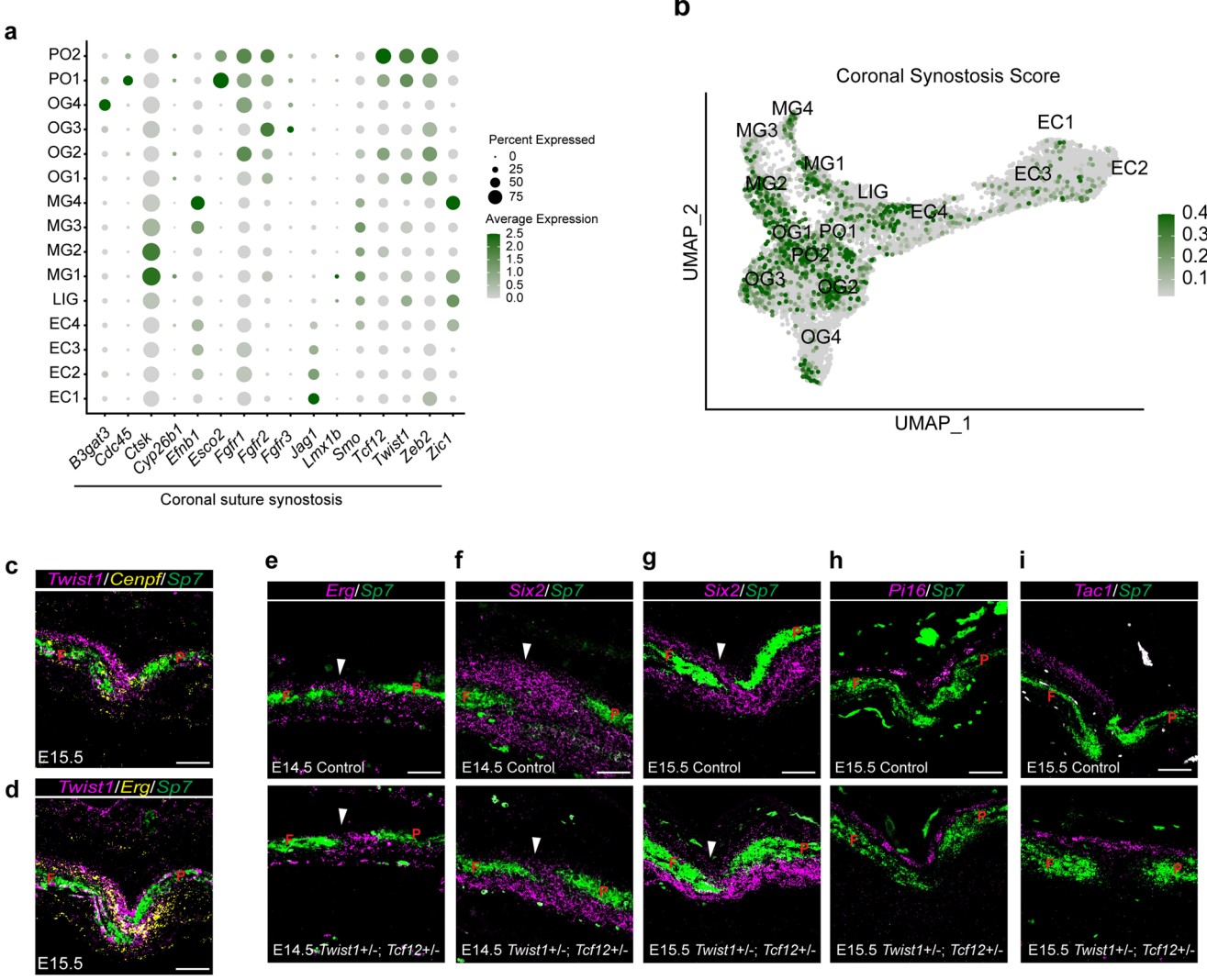

**Fig. 7 Enrichment and requirements of coronal synostosis genes for osteogenic cells. a** Dot plot of genes associated with coronal suture synostosis. The size and color intensity of the dots correspond respectively to the percentage of cells within each cluster expressing the indicated gene, and the average expression level. **b** Module score for coronal synostosis genes plotted on to the osteogenic/mesenchymal subset UMAP. **c–i** Combinatorial in situ analysis of coronal sutures for indicated markers at E14.5 or E15.5. *Sp7* marks the frontal (F) and parietal (P) bones. **c, d** *Twist1* expression overlaps with markers for PO1 (*Cenpf*) and OG1 (*Erg*) within coronal suture mesenchyme of wild-type E15.5 mice. **e–i**, expression of OG1 markers *Erg* and *Six2* is lost in the suture mesenchyme (arrowheads) of *Twist1*[+/−]; *Tcf12*[+/−] mice, yet EC1-3 marker *Pi16* and LIG marker *Tac1* are unaffected. In situs were performed in biological triplicate. Scale bars = 50 μm; images in (**c–i**) are at the same magnification.

distinguished by expression of several chondrogenic markers, including *Matn4*, *Ctgf*, and lower levels of *Col2a1* and *Acan*. Cartilage formation beneath sutures has been linked to normal and pathological suture fusion[62,63], yet the precise source of such cartilage has remained unclear. It will be interesting to assess whether the periosteal dura layers contribute to the natural and ectopic cartilage formation associated with suture closure. In addition, the periosteal dura layers express multiple signaling factors implicated in suture regulation, including *Tgfb2*, *Fgf2*, *Gdf10*, and *Ctgf*[64], and CellPhoneDB analysis points to signaling interactions between the periosteal dura layers and preosteoblasts. Our single-cell atlas therefore provides testable hypotheses for how specific meningeal cell types may regulate cranial bone formation in a paracrine fashion.

The ectocranial mesenchyme above the coronal suture is also known to regulate suture patency[7,22], and here we identify at least four distinct layers. The two outermost layers likely represent the lower limits of the skin, including dermal reticular fibroblasts

(*C1qtnf3*[+]) and non-osteogenic *Ly6a*[+] (*Sca1*[+]) hypodermis[31,65,66]. Ectocranial expression of *Jag1* and *Epha4* has been implicated in coronal suture formation[7,22], and we find *Jag1* expression in the hypodermis and *Epha4* in the dermal reticular fibroblast layer. Thus, multiple ectocranial layers likely play roles in suture regulation. Below the hypodermis, we identified a *C1qtnf3*[+]/*Tnmd*[+] layer that we termed suprasutural, as well as a related and tightly associated layer enriched for expression of genes involved in ligament formation, cellular contraction, and mechanosensation. By analyzing the tendon/ligament reporter *Scx:GFP*, we find that a ligament persists in this location until at least 10 months after birth. One possibility is that this ligament, which connects the lateral tips of the frontal and parietal bones, contributes to the flexibility of the sutures, for example to accommodate compression of the calvarium during birth. More speculatively, this population could also function to interpret mechanical forces and transmit these to the osteogenic cells within the suture, thus coupling expansion of the brain to cranial growth. Consistently, CellPhoneDB predicts strong ligand-

receptor interactions between the ligament-like population and osteogenic cells, including Tgfβ, Fgf, Wnt, and Igf1 signaling families.

The broad distribution of craniosynostosis-related genes within our dataset suggests diverse etiologies of synostosis. However, over half of coronal synostosis genes had enriched expression within osteogenic clusters, including Twist1 and Tcf12 that have been implicated in negative regulation of the rate of bone growth in the cranium[20]. It will be interesting to test whether this reflects more general roles of coronal synostosis genes in regulating osteogenesis, although other coronal synostosis genes had pre-ferential enrichment in meningeal (Ctsk, Efnb1) or ectocranial (Jag1) layers. It is possible that differences in gene expression, and/or the principal cell populations and processes they control, underlie the distinct genetic sensitivities of particular cranial sutures. One prominent unique feature of the overlapping coronal suture is the asymmetric distribution of osteoprogenitors, which is not observed at the non-overlapping frontal suture. Indeed, our comparison of the coronal and frontal sutures reveals that, whereas the cell types themselves are largely shared, their topo-logical arrangements and gene expression profiles are suture-specific. Future work will be needed to understand how the geometry of each suture is determined, and how this impacts the sensitivity of particular sutures to loss in distinct human cra-niosynostosis syndromes.

## Methods

**Animals**. Mouse work was performed in accordance with UK Home Office reg-ulations under approved project licences and the University of Southern California Institutional Animal Care and Use Committee (USC IACUC Protocol #20552). Mice were housed in cages with up to five adults or two adults with a litter. Genotyping of the following mouse lines used the primers listed in Supplementary Table 1: Tcf12 and Twist1[9], Scx-GFP[67], Six2-Cre[68] (https://www.jax.org/strain/009606), Six2-CreERT2[68] (https://www.jax.org/strain/032438), Ai9 (https://www.jax.org/strain/007909)[69],R26-CAG-LSL-Sun1-sfGFP-myc (https://www.jax.org/strain/021039)[70]. For Six2-Cre experiments, timed matings between Six2-Cre heterozygous males and Ai9 homozygous females were performed, and mothers were sacrificed at E16.5 to harvest embryos. To harvest postnatal time points for Six2-Cre;Ai9 mice or Scx-GFP mice, genotyped animals were sacrificed at 3 weeks. For tamoxifen treatments, newborn pups (P0) were obtained from crosses of Six2-CreERT2; R26-CAG-LSL-Sun1-sfGFP-myc males and Six2f/f females, and injected with 10 uL of a 20 mg/mL tamoxifen solution dissolved in corn oil into the abdominal cavity. Swiss Webster foster mothers were used to avoid maternal rejection, and pups were sacrificed at P7.

**Coronal suture dissociations**. E15.5 and E17.5 embryos were isolated and the bony skull was dissected away from the skin and brain. For E15.5 embryos the coronal suture was dissected away from the skull cap using microdissection scis-sors, and 10 pooled coronal sutures were rinsed in PBS and enzymatically dis-sociated with a final concentration of 3 mg/mL of Collagenase II (Worthington) and 4 units/mL of Dispase (Corning) in DMEM/F12 (Corning) for 45 min. For E17.5 embryos coronal sutures were dissected out in ice-cold PBS using a scalpel blade, isolating a strip containing the overlapping frontal and parietal bone fronts (which appear opaque compared to adjacent regions) and avoiding the most apical and basal aspects of the suture. Isolated sutural strips from embryos from two litters in 3 batches (batch 1, 10 sutures from litter 1; batch 2 and 3, 3 sutures each from litter 2) were cut into small fragments in HBBS and digested using Col-lagenase IV (Worthington, USA; final concentration in HBBS of 2 mg/mL) for 30 min. Dissociation was terminated with 2% Fetal Bovine Serum and cells were passed through a 0.35 μM filter (E15.5) or Pluri-strainer Mini 70 μm (E17.5; pluriSelect Life Science, Germany). For E15.5 sample preparation, dead cells were removed using the Dead Cell Removal kit (Miltenyi Biotec 130-090-101) and cells counts were determined with a hemocytometer. The three batches of E17.5 dis-sociated cells were separately sorted by FACS to remove debris, cell doublets and likely dead cells (BD FACSAria Fusion; 100 μM nozzle; Supplementary Fig. 13) prior to library preparation.

**scRNA-seq library preparation and sequencing**. Transcriptome libraries for single cells were captured using 10X genomics Chromium Single Cell 3' Library and Gel Bead Kit v2 following manufacturer's guidelines. Sequencing for E15.5 coronal sutures was performed with Illumina's HiSeq 3000/4000 PE Cluster Kit at the Children's Hospital Los Angeles Molecular Genomics Core, and for E17.5 cells, PE sequencing was run on an Illumina HiSeq 4000 at the Oxford Genomics Centre

Wellcome Centre for Human Genetics, Oxford, achieving an average of ~150,000 mean reads per cell for E15.5 and ~92,000 mean reads per cell for E17.5.

**Bioinformatics analysis**. Quality control of raw reads was performed with FastQC version 0.11.7, fastq_screen version 0.7.0 and multiqc[71] version 0.9. Samples were counted individually and aggregated with cellranger (10X Genomics) version 2.1.1 using the mm10 mouse transcriptome. All other parameters were set to their default values. Data analysis was performed with Seurat[23] version 3.2.0. The aggregate gene/barcode matrix ('raw_gene_bc_matrices_mex') was loaded using 'CreateSeuratObject' and 'Read10X' with 'min.cells = 10, min.genes = 200'. Cells were filtered to exclude cells with fewer than 1000 UMIs, fewer than 1000 genes, more than 7.5% mitochondrial content (calculated as the fraction of reads assigned to a gene on the chromosome 'MT'), and more than 3% Hbb/Hba content (to remove red blood cells, and cells with high contamination for red blood cell-specific genes). Filtered cells were normalized with SCTransform, and cell-cycle scoring and regression were performed for each dataset with the default list of human cell-cycle genes (converted to mouse gene symbols). Datasets were inte-grated based on the tutorial "Integration and Label Transfer" from Seurat (https://satijalab.org/seurat/archive/v3.0/integration.html). In brief, an object list including both datasets was created, 3000 features were selected using the SelectInte-grationFeatures following by PrepSCTIntegration, anchors were identified using FindIntegrationAnchors, and the data were integrated using IntegrateData. Dimensionality reduction was performed using the first 30 principal components (PCs) with UMAP[72] (min_dist = 0.5, n_neighbors = 30). Clustering was per-formed with 'FindClusters' and 'resolution=1'. Cluster marker genes were identi-fied with 'FindAllMarkers' using parameters 'min.pct = 0.2, logfc.threshold = 0.5, max.cells.per.ident = 1000, min.cells.gene = 5'. Secondary dimensionality reduc-tion and clustering of the osteogenic and mesenchymal subset were performed as above, after subsetting for cluster numbers 0, 1, 2, 4, 5, 7, and 9. The new subset was reclustered at 'resolution=0.6'. Trajectory analysis was carried out on the osteogenic subset (clusters OG1-4, PO1-2) using Monocle 3[73]. The Seurat inte-grated object was converted into a Monocle cell dataset and the cluster information and UMAP coordinates carried over from the Seurat object prior to following the Monocle3 recommended protocols and parameters (from the learn the trajectory graph section onwards at https://cole-trapnell-lab.github.io/monocle3/docs/trajectories/#learn-graph). The trajectory root was manually selected, using known osteogenic pathway gene expression patterns (shown in Fig. 5c) working back from the most differentiated cells, to inform the likely starting point of the lineage. Cell–cell communication analysis was performed using CellphoneDB v2.1.4[50], after transforming mouse genes to human homologs. We prioritized potential interac-tions (p-value < 0.01) and manually selected those that were of biological relevance. Synostosis genes for the coronal module were determined based on published human craniosynostosis reports[52,55]. Scores were calculated using the AddModu-leScore function in Seurat: https://rdrr.io/github/satijalab/seurat/man/AddModuleScore.html. Module scores were calculated using the RNA assay and control features were set to 10. FeaturePlots to visualize the module score had a minimal cutoff of q10 and a maximal cutoff of q90. For statistical analysis, module scores were extracted for each cell, and cells were group by region (Ectocranial: EC1-4 and LIG, Osteogenic: OG1-4 and PO1-2, Meningeal: MG1-4). Values less than zero were set to zero to replicate the data visualized in the FeaturePlots. Statistical significance was determined using a pairwise Wilcoxon test with Ben-jamini & Hochberg correction in R. All included DotPlots reflect z-score scaled data generated from the DotPlot() function that enables visualization of lowly expressing clusters with highly expressing clusters on the same scale (https://rdrr.io/bioc/scater/man/plotDots.html). Dot size was determined by setting the para-meters dot.min = 0 and dot.scale = 9. Expression scales were determined by setting the parameters minimal scale col.min = 0 and maximum scale col.max = 2.5. All FeaturePlots and VlnPlots for gene expression were generated from the RNA assay expression values, with a minimal cutoff of q10 and a maximal cutoff of q90 for FeaturePlots.

**RNAScope**. RNAscope in situ hybridization was performed using the RNAscope Multiplex Fluorescent Kit v2 (Advanced Cell Diagnostics, Newark, CA) according to the manufacturer's protocol for fixed-frozen sections, with the modifica-tion that to retain optimal sectioning quality the heat antigen retrieval was omitted. TSA® Plus (Fluorescein, Cy3 and. Cy5) reagents were used at 1:1000.

**Immunofluorescence and staining on cryosections**. Wild-type C57BL/6 embryos were collected at E16.5, rinsed in ice-cold PBS followed by head dissec-tion, overnight incubation in 4% PFA in PBS at 4 °C, and processing through a sucrose gradient. 12 μM sections were prepared, air dried, and blocked for 1 hour in 5% goat serum in PBST. Rabbit anti-Sp7 (1:750; sc-22536-R, Santa Cruz Bio-technology) and Chicken anti-mCherry (1:500, NBP2-25158, Novus Biologicals) were added overnight in 5% goat serum in PBST. Slides were washed and stained with Alexa Fluor 488 and 647 secondary antibodies (all 1:500; Thermo Fisher Scientific). Adult samples were collected and the skin was removed from the skull. Heads were fixed overnight and decalcified in EDTA for up to 2 weeks, followed by processing through a sucrose gradient. Embedded heads were sectioned at 12 μM and air-dried slides were washed in PBS and stained with DAPI.

**Immunohistochemistry on parrafin sections**. Wild-type C57BL/6 embryos were collected at E17.5 and rinsed in ice-cold PBS for 30 minutes followed by head dissection, skin removal, and overnight incubation in 4% PFA at room temperature. Heads were then washed multiple times in PBS, decalcified for 2 hours in Calci-ClearTM Rapid (HS-105, National Diagnostics), dehydrated, and paraffin embedded. Embedded tissue was sectioned (5 µm) and then rehydrated, stained, and visualized using ImmPRESS® HRP Anti-Rabbit IgG (Peroxidase) Polymer Detection Kit (MP-7451, Vector Laboratories). Primary antibodies were diluted in cold TBS and incubated overnight at 4 °C. In order to detect primary antibodies other than those raised in rabbit, donkey anti-sheep/goat/rat IgG-HRP was used (all 1:200, A16041, sc-2020, A18739, respectively). The tissue sections were subsequently imaged using an Olympus BX60 Microscope (Olympus) and/or Nano-Zoomer 2.0 HT (Hamamatsu). Secondary antibodies include donkey anti-sheep IgG-HRP (A16041), donkey anti-goat IgG-HRP (sc-2020), and donkey anti-rat IgG-HRP (A18739). To study Tcf7/Cd200 and Tcf7/Dmp1 localization, double immunofluorescence staining was performed on 5 µm E17.5 sections. Following deparaffinisation, rehydration, and heat-mediated antigen retrieval in 10 mM sodium citrate buffer solution (pH 6), samples were blocked in 4% Donkey Serum (D9663, Sigma-Aldrich) for 30 min. Individual sections were then incubated overnight at 4 °C with a mixture of Tcf7 (1:200; C63D9, Cell Signaling Technology) and Cd200 (1:200; AF2724, R&D systems) or Tcf7 (1:200; C63D9, Cell Signaling Technology) and Dmp1 (1:400; AF4386, R&D systems) primary antibodies. Antigen detection was performed using the appropriate combination of the Alexa Fluor 488, 555, and 647 secondary antibodies (all 1:500; A21206 or A11015, A21432, A31573; Thermo Fisher Scientific) for 2 h at room temperature in the dark. All primary/secondary antibodies were diluted in SignalBoost™ Immunoreaction Enhancer Kit (407207–1KIT, Calbiochem). After three washes in PBS, sections were incubated with DAPI (1 µg/mL) (Roche, cat # 10 236 276 001). Following multiple washes in PBS, slides were mounted using Vectashield® Antifade Mounting Medium (H-1000Vector Laboratories, Inc.). Imaging was performed using a Zeiss LSM 780 Upright Multi-Photon Confocal Microscope with LD LCI PA 25×/0.8 DIC WD = 0.57 mm Imm Corr (UV)VIS-IR (Oil-Immersion) and Plan-Apochromat 63x/1.4 Oil objectives. Images were obtained using ZEISS ZEN Microscope software. When using two rabbit antibodies, for example the co-localization of Gja1 (1:100; 3512, Cell Signalling Technology) and Crabp2 (1:750; 10225-1-AP, Proteintech), we used the TSA Cyanine 3 Plus Evaluation Kit (NEL744E001KT, Perkin Elmer) following the manufacturers' instructions. Primary antibodies were visualized using ImmPRESS® HRP Anti-Rabbit IgG (MP-7451, Vector Laboratories) followed by incubation with Fluorescein (FP1168) or Cyanine 3 (FP1170) Amplification Reagents (both 1:200) and imaged by a Zeiss LSM 780 Upright Multi-Photon Confocal Microscope (same parameters as above).

**Reporting summary**. Further information on research design is available in the Nature Research Reporting Summary linked to this article.

## Data availability

The RNAseq data generated in this study, including raw and processed files, are available in the GEO repository, accession: "GSE163693". All other data are included in the article and Supplementary Information or available from the corresponding authors upon reasonable request.

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

## Acknowledgements

The authors thank Kevin Clarke and Craig Waugh from the WIMM Flow Cytometry Facility, and Claire Arata, Maxwell Serowoky, and Francesca Mariani from the University of Southern California for help with single-cell dissociation and isolation. We thank Julie Siegenthaler from the University of Colorado Anschutz Medical Campus for helpful advice and feedback. We thank the Oxford Genomics Centre at the Wellcome Centre for Human Genetics and the Children's Hospital Los Angeles Molecular Pathology Genomics Core for next-generation sequencing. Work was supported by Wellcome (102731 to A.O.M.W.), Action Medical Research (GN2483 to S.R.F.T.), VTCT Foundation Fellowship (S.R.F.T., A.O.M.W.), the MRC through the WIMM Strategic Alliance (G0902418 and MC_UU_12025), Burroughs Wellcome Trust (D.T.F.), HHMI Hanna H. Gray Fellows Program (DTF), and National Institutes of Health (R01DE026339 to J.G.C. and R.E.M.). For the purpose of open access, the authors have applied a CC BY public copyright license to any author-accepted manuscript version arising from this submission. The views expressed in this publication are those of the authors and not necessarily those of funding sources.

## Author contributions

D.T.F., A.O.M.W., R.E.M., S.R.F.T., and J.G.C. conceived and designed the study. D.T.F., Y.Z.N.K., N.A., G.W., K.C.T., and S.R.F.T. carried out the single-cell and bioinformatic analysis. D.T.F. performed the RNAScope validation experiments, and H.M. the immunolocalizations. D.T.F., H.J.C., H.B., and R.P. performed transgenic mouse experiments, with A.E.M. contributing and advising on the Scx-GFP mouse work. S.R.F.T., A.O.M.W., and J.G.C. supervised the research. D.T.F., S.R.F.T., and J.G.C. wrote the paper with contributions from all authors.

## Competing interests

The authors declare no competing interests.
