## [Peer Review File · Nature Communications]

Reviewers' Comments:

Reviewer #1:

Remarks to the Author:

Thank you very much for providing me with the opportunity to review the manuscript entitled "The developing mouse coronal suture at single-cell resolution" for Nature Communications. Here, the authors sought to generate a single-cell atlas to describe better developing coronal suture in mice. Single-cell transcriptomes generated and analyzed. Key findings of this study include the following: Identification of an Erg/Pthlh progenitor that may give rise to two pre-osteoblast pathways – along the bone tips/suture and periosteal tissues.

Populations of cells with chondrocyte differentiation potential within the dura.

A ligament-like population of cells connecting the frontal and parietal plates.

Differential expression of genes associated with coronal and midline (metopic and sagittal) synostosis within multiple tissue types.

These results only confirm the complex nature of the developing coronal suture and may potentially act as a framework by which to further investigate both monogenic and "sporadic" synostosis. While the study does provide an impactful contribution to the literature, it is purely descriptive in nature. I have a few considerations as several questions remain unanswered (and perhaps answered with follow-up studies). Please see both major and minor comments below.

Major Comments:

1. An imbalance between proliferation and differentiation of osteogenic cells at the osteogenic fronts and mesenchyme has long been thought to lead to suture synostosis, particularly coronal synostosis. The data presented here does seem to support that, but the authors did not investigate this any further. Where proliferative studies performed? The authors should consider using numerous mouse models available for syndromic coronal synostosis to provide context to the identified osteogenic/mesenchymal cell subsets.

2. While the authors provide spatial data, temporal data is lacking, which would complement the study. Only two time points were chosen: E15.5 and E17.5. Coronal suture development does begin early (< E12.5) and continues postnatally until the completion of brain growth. Were earlier and later time points evaluated? Authors should consider evaluating the P3/P7 and P14/P21 coronal suture sets to justify the paper's title.

Minor comment:

One particularly interesting finding is the ectocranial cluster identified at E15.5. The methodologies, as described, seemed to limit this finding. The sutures with both a small segment of frontal and parietal bones were removed. These were not full-thickness sections, correct? As described, the overlying thin periosteal/dermal/epidermal were presumably stripped (perhaps inadvertently). Please provide commentary on this.

Reviewer #2:

Remarks to the Author:

Thank you for allowing me to review the manuscript entitled "The developing mouse coronal suture at single-cell resolution" by Farmer et al. This is an original investigation assessing the cellular diversity in the developing coronal suture in mice by single cell discovery sequencing complemented with spatial transcript identification.

The experiments presented here offer new information regarding the cellular identity and transcriptome of the developing skull ectocranial tissues and meninges, relevant for researchers in the specific field. The paper offer insight into the location, identity of osseous and cartilaginous precursors, while giving sense to their special arrangement. Because the experiments are limited to discovery sequencing, supported by immunofluorescence or RNAscope, conclusions regarding the functional properties of the cells and genes identified cannot be sustained, but need further mechanistical biology to be claimed as such. The conclusion of the possible roles of cells or genes

in the pathogenesis of cranioostenosis is also not sustained by these results and would also need a completely new set of experiments to be proven. The last statement in the abstract "This single-cell atlas provides a resource for understanding development of the coronal suture..." accurately reflects what this paper is about. Any higher claim or interpretation of the presented results requires a mechanistic validation.

I believe that in order to warrant interest to a broader audience either the disease coupling part of the manuscript would have to be substantially strengthened. As is now this part is too superficial and therefore I suggest a more specialized journal for publication.

Specific comments:

Introduction

Line 44, cranial and calvarial are synonyms, please re structure the sentence to make it clearer for a non-expert reader.

Line 84, the expression of Twist1 and TCF12 in the dural and ectocranial tissues does not imply a multiple origin of craniosynostosis beyond a broad expression of these two transcription factors. Were these overexpressed regarding other transcription factors relevant for mesenchymal development?

Results

The presence of a major inter dural veins along sutures should be discussed. Also, one issue in general with the paper is that the authors should provide the reader the opportunity to judge better what is "suture-specific". Perhaps showing larger portions of non-suture bones? Or by whole mount stainings. The cartoon in figure 4j suggests it is only OG2 that is not generally positioned in or around cranial bone. This pertains to entire Figure 4 – would benefit from more zoomed out images. Similarly, I find it hard to judge the non-symmetric distribution of Erg and Pthlh without seeing further away from the suture for comparison.

Line 185 I do believe that claiming that substance P is indicative of tendon mechanoreception is narrowing the broad spectrum of functions linked to this neuropeptide. Furthermore, the finding that the same cells also express SMA genes is confusing. Perhaps the authors should try to re-cluster this population separately to see whether putative ligament and smooth muscle cells separate. Could it be that these SMA-gene expressing cells actually are derived from the blood vessels? As this is written now with the two examples Tac1 and Chodl the reader is led to the conclusion that all the information points towards a tendon being present there which is an interesting finding that makes intuitive sense, but the authors should then substantiate this by showing this in tissue sections at a later developmental timepoint.

The presentation on osteoblasts is accurate and informative and, in my view, best represents an unbiased and broad view of the findings. Where is Sp7 expression in the transcriptomic analysis? This is a marker used throughout the paper in most figures and the authors use it in the final identification of OG2 as the tip-specific pre-osteoblast. Right now, it is only shown in supplementary figure but I would like to see it plotted in figure 4.

Line 270. Text references fig 6c when it probably should be 5c. In the figure the legend with the color code lacks description (only numbers now) – please add what the numbers depict in the figure for increased readability.

The section on "signaling interaction" is the weakest part of the manuscript. The authors have not studied the interaction between any of the mentioned molecules but speculate of their existence based on location of the cells that secrete them, a too superficial analysis based on numbers of potential interactions and prior knowledge. Given that one strong interaction could be enough for biological significance it is weird to make this statement based on number of potential interactions only. This argument would be somewhat strengthened if the authors at least would normalize for expression of possible interactions (i.e., perhaps MG and EC cells simply express more secreted

factors in general?). Due to the superficial nature of this analysis, I believe that this part could be removed without lessening the impact of the study.

Figure 6: For figure 6a the authors should provide methods section explaining how 75% of cells can have 0.1 expression in light of that 10x provides UMIs. I guess this is because the authors are using data normalized in some way, but this should be clearly explained. Also seems like equally strong expression in midline as in coronal suture synostosis – main difference is percentage of cells expressing each gene. It is not clear beyond slightly better detection that difference in disease could be driven by these small differences. Given that 10x have quite low coverage is it possible that many negative cells represent false negatives in the midline suture genes. Could this be statistically modelled?

With regards to fig 6b I think the authors have made some weird choices given that many of the monogenetic causes are autosomal dominant. Why is an average calculated? Would it not be better to add signal on top of each other – expression of one mutated gene could be enough in an autosomal dominant disease.

For disease coupling the authors are also missing a golden opportunity to compare an independent data set from Holmes et al. Now they have hidden this away in supplementary figure 11 and only performed a very superficial analysis. I would suggest comparing not only midline vs coronal genes but also cell types from the two data sets.

Discussion

Line 298, the first paragraph of the discussion states that single cell sequencing bridges the current state of knowledge to provide answers on craniosynostosis. Although discovery sequencing is, in effect, and as shown by the authors, useful to know the parts list, it is insufficient to understand the pathogenesis at this point. I would focus on the presented results more as a step, mainly useful for researchers in the field, and less as the answer of what goes wrong in craniosynostosis, which this paper doesn't.

Line 354: the claim that the expression of genes present in ligaments means that the ectocranial tissue acts like one is not justified and should be changed. Even if ligament or ligament like tissue was actually found that does not mean that it fulfills any tension bearing function. Ligaments are a common occurrence within the cranium and fulfill no such a function. These types of possibly or probable claim, in my view, weaken the paper. The authors provide an extraordinarily valuable resource for the research community, claims on the function on the different genes found should be supported on the available results.

Methods

The authors pooled 10 different mice, was each individual identified for analysis? Were there any outliers?

Reviewer #3:

Remarks to the Author:

Farmer et al present a single-cell RNA sequencing study of the coronal suture in the mouse skull, a structure that is ablated in human disease and is responsible for normal bone growth and repair. The authors combine bioinformatic analyses to parse distinct cell fate states and combinatorial in situ staining to locate these cell states within the tissue at E15.5 and E17.5. Rather than focus only on the cell types within the suture, between frontal and parietal bones, the authors make an important additional contribution by first classifying subtypes of meningeal cells that reside below the suture and ectocranial mesenchyme that will give rise to dermis above the suture. Restricted and layered expression of uniquely represented markers provides the first insight into cell fate diversity and architecture within these tissues so early in suture development. It is exciting then, that the authors find a ligamentous population capping the suture joint that has not been previously described. The authors then perform more detailed bioinformatic analysis of the

osteoblast lineages in their data set and find not only novel markers of early osteoblast progenitors but, significantly, their data suggest two types, or rather lineages, of osteoblasts residing either in the bone tips or edges of the bone center. Further, they provide a matrix of ligand and receptor expression for multiple pathways across osteoblast lineages adding clarity to the debated presence of signalling molecules, such as Indian Hedgehog. The authors end by integrating genes known to contribute to disease in the coronal suture versus the midline sutures with their data set to highlight similarity and differences in molecular regulation of a debilitating disorder – craniosynostosis, which is important for understanding suture biology generally. Taken together, this article adds an important resource for the field and provides a basis for mechanistic predictions not only relevant to the developmental biologist but clinicians.

This article is appropriate for publication in Nature Communication as it an insightful, well-presented and important contribution, however, minor comments remain.

1. The images presented feature quite a lot of overexposed signal throughout. The saturated signal makes it tricky to appreciate the true extent of signal overlap and size of RNA loci. This is especially true for fig. 3. H, where a key point is that Podn1 is expressed within the bone edge but it is not clear whether these cells are also Sp7 positive.

2. The figure legends could benefit from more detail, e.g. Fig.4. where a is listed simply as lineage analysis. For the uninitiated trainee, it might be helpful to point out that the grey line connects relationships between cell fates.

3.Indeed, it may be helpful to further detail the parameters used in both Seurat clustering and the monocle based lineage analysis. While this reviewer understands there are recommendations within these packages, there is little procedural information on how the resolution of these clusters was chosen as an example. As these parameters can especially influence lineage associations (e.g. Fig 4.), more detail is advised.

4. As the numbers below the x axis in fig 4. c are too small to see and the legend includes that information, these could be deleted.

Re NCOMMS-20-50899

We were pleased to see that all three reviewers were largely positive and appreciated that our single-cell study of the coronal suture provides a valuable resource for the research community. Here we respond to the reviewers' comments, point-by-point, with our rebuttal shown in blue text.

Reviewers' comments:

Reviewer #1 (Remarks to the Author):

These results only confirm the complex nature of the developing coronal suture and may potentially act as a framework by which to further investigate both monogenic and "sporadic" synostosis. While the study does provide an impactful contribution to the literature, it is purely descriptive in nature.

We agree that the original version of our study was largely descriptive. We have now added key mechanistic data from what was originally intended as a follow-up study. As described below in more detail, this includes lineage-tracing of the osteoprogenitor population using *Six2-Cre* and *Six2-CreERT2* mice, and analysis of changes to multiple suture populations in a craniosynostosis mouse mutant.

Major Comments:

1. An imbalance between proliferation and differentiation of osteogenic cells at the osteogenic fronts and mesenchyme has long been thought to lead to suture synostosis, particularly coronal synostosis. The data presented here does seem to support that, but the authors did not investigate this any further. Were proliferative studies performed? The authors should consider using numerous mouse models available for syndromic coronal synostosis to provide context to the identified osteogenic/mesenchymal cell subsets.

As suggested, we have now analyzed coronal suture subsets in a *Twist1/Tcf12* mouse model of Saethre-Chotzen Syndrome, one of the most common form of syndromic coronal synostosis (new Fig. 7). In particular, we find severe reduction of progenitor markers *Six2* and *Erg* at multiple embryonic stages, yet no effect on ectocranial expression of *Pi16* or *Tac1*. In our previous study of these mutants (Teng et al., 2018), we had performed proliferative studies in these mutants that showed an increase in proliferative pre-osteoblasts, and hence we have not repeated these experiments here. By identifying the embryonic suture progenitors affected in *Twist1/Tcf12* mutant mice, we have for the first time revealed the precise cellular etiology of suture fusion in this model. More broadly, this example shows the power of our single-cell resource for identifying the cell types affected in different forms of craniosynostosis.

2. While the authors provide spatial data, temporal data is lacking, which would complement the study. Only two time points were chosen: E15.5 and E17.5. Coronal suture development does begin early (< E12.5) and continues postnatally until the completion of brain growth. Were earlier and later time points evaluated? Authors should consider evaluating the P3/P7 and P14/P21 coronal suture sets to justify the paper's title.

We have now expanded our analysis to include a wider range of embryonic and postnatal timepoints. In new Fig. 7f,g, we show that *Erg* and *Six2* mark suture progenitors earlier at E14.5. In new Fig. 5l-n, we use *Six2-Cre* and *Six2-CreERT2* lineage tracing to show contribution of *Six2*⁺ cells to the coronal suture mesenchyme at P7 and P21. In new Fig. 4e-g, we use a *Scx:GFP* reporter to show persistence of the ligament-like population (LIG) into adulthood (3 weeks, 3 months, and 10 months after birth). These new findings show that the *Six2*⁺ progenitor population arises as early as E14.5 and persists until at least 3 weeks after birth, and that the embryonic ligament-like population is a permanent aspect of the coronal suture (see also response to reviewer 2). These new experiments have also led to the addition of several co-authors: Helena Bulgacov and Riana Parvez from Andrew McMahon's lab for the *Six2-Cre* and *Six2-CreERT2* mice, and Amy Merrill for the *Scx-GFP* mice.

Minor comment:

One particularly interesting finding is the ectocranial cluster identified at E15.5. The methodologies, as described, seemed to limit this finding. The sutures with both a small segment of frontal and parietal bones were removed. These were not full-thickness sections, correct? As described, the overlying thin periosteal/dermal/epidermal were presumably stripped (perhaps inadvertently). Please provide commentary on this.

We assume the reviewer is referring to cluster EC3, which had higher representation at E15.5 than E17.5. However, all ectocranial clusters were recovered at both E15.5 and E17.5, and our new comparison with a previously published frontal suture dataset shows that clusters EC1-3 in our study are the same as the hypodermal HD cluster recovered in the frontal suture study. It thus appears that dissections from independent labs do not generally remove these layers. That said, we do provide commentary in the Results that the outer-most *C1qtnf3*⁺ ectocranial layer that expresses *Epha4* does appear to have been lost following dissection.

Reviewer #2 (Remarks to the Author):

The paper offers insight into the location, identity of osseous and cartilaginous precursors, while giving sense to their special arrangement. Because the experiments are limited to discovery sequencing, supported by immunofluorescence or RNAscope, conclusions regarding the functional properties of the cells and genes identified cannot be sustained, but need further mechanistical biology to be claimed as such. The conclusion of the possible roles of cells or genes in the pathogenesis of

craniosynostosis is also not sustained by these results and would also need a completely new set of experiments to be proven. Any higher claim or interpretation of the presented results requires a mechanistic validation. I believe that in order to warrant interest to a broader audience either the disease coupling part of the manuscript would have to be substantially strengthened. As is now this part is too superficial and therefore I suggest a more specialized journal for publication.

We agree with the reviewer on the need for mechanistic biology/validation to appeal to a broader audience. We have therefore included 3.5 figures of new data that provide significant insights into the functional properties of the osteogenic cells and their disruption in Saethre-Chotzen Syndrome, a common monogenic form of coronal craniosynostosis. In revised Fig. 5, we now use constitutive *Six2-Cre* and conditional *Six2-CreERT2* lines (in collaboration with members of Andrew McMahon's lab) to demonstrate that the OG1 progenitor population gives rise to both new osteoblasts and postnatal suture mesenchyme. This is an important finding as it points to OG1 osteoprogenitors being the embryonic cells that give rise to the better understood postnatal sutural stem cells. In a new Fig. 6, we have also analysed changes to coronal suture cell clusters in a *Twist1/Tcf12* mouse model of Saethre-Chotzen Syndrome. We find a specific loss of OG1 progenitors in these mutants, with ectocranial clusters being largely unaffected. These new findings reveal for the first time the identity of the embryonic progenitors affected in this disease. Together, we feel that the inclusion of substantial new lineage tracing and disease modelling data greatly broaden the impact of our study.

Specific comments:

Introduction

Line 44, cranial and calvarial are synonyms, please re structure the sentence to make it clearer for a non-expert reader.

Cranial or cranium is now used in place of calvarial throughout the document.

Line 84, the expression of *Twist1* and *TCF12* in the dural and ectocranial tissues does not imply a multiple origin of craniosynostosis beyond a broad expression of these two transcription factors. Were these overexpressed regarding other transcription factors relevant for mesenchymal development?

We agree that this was confusing and have modified this section to better focus on the fact that 9/16 coronal synostosis genes have particular enrichment in osteogenic clusters. In revised Fig. 7d,e, we show that *Twist1* expression overlaps primarily with the OG1 marker *Erg* and the proliferative progenitor marker *Cenpf*, in agreement with our new data showing loss of OG1 markers *Erg* and *Six2* in *Twist1/Tcf12* mutants.

Results

The presence of a major inter dural veins along sutures should be discussed. Also, one issue in general with the paper is that the authors should provide the reader the

opportunity to judge better what is “suture-specific”. Perhaps showing larger portions of non-suture bones? Or by whole mount stainings. The cartoon in figure 4j suggests it is only OG2 that is not generally positioned in or around cranial bone. This pertains to entire Figure 4 – would benefit from more zoomed out images. Similarly, I find it hard to judge the non-symmetric distribution of *Erg* and *Pthlh* without seeing further away from the suture for comparison.

We note the likely presence of major inter dural veins in Supplementary Figure 3c and discuss this in the main text: “*Rgs5+* pericytes were interspersed with *Gjb6+* cells in the MG4 arachnoid layer, consistent with the prominent vasculature extending from the border of the dura mater and through the arachnoid mater to the pia mater²⁶ (Supplementary Fig. 3c).”

Several existing and new zoomed out images show the restriction of OG1, OG2, and LIG clusters to the coronal suture. Existing E15.5 (Fig. 5i) and new E14.5 (Fig. 7f) images of *Erg* show lack of expression away from the suture. In new Fig. 4, *Tac1* in situ at E15.5 and analysis of *Scx-GFP* at 3 weeks, 3 months, and 10 months after birth clearly show restriction of the ligament-like population to above the coronal suture.

Line 185 I do believe that claiming that substance P is indicative of tendon mechanoreception is narrowing the broad spectrum of functions linked to this neuropeptide.

We have modified description of Substance P to clarify its broad spectrum of functions: “*Tac1* encodes four neuropeptides including Substance P, a neuropeptide with diverse biological functions that has also been implicated in tendon mechanosensation³².”

Furthermore, the finding that the same cells also express SMA genes is confusing. Perhaps the authors should try to re-cluster this population separately to see whether putative ligament and smooth muscle cells separate. Could it be that these SMA-gene expressing cells actually are derived from the blood vessels? As this is written now with the two examples *Tac1* and *Chodl* the reader is led to the conclusion that all the information points towards a tendon being present there which is an interesting finding that makes intuitive sense, but the authors should then substantiate this by showing this in tissue sections at a later developmental timepoint.

In Supplementary Figure 5b,c, we have now performed re-clustering of the LIG population and find that the ligament and smooth muscle cells do in fact separate. In particular, *Acta2+* smooth muscle cells form a distinct subcluster from ligament cells expressing *Tac1*. We have also used *Scx-GFP* reporter mice (in collaboration with the lab of Amy Merrill) to demonstrate the persistence of this ligament population up to 10 months of age (Fig. 4e-g).

The presentation on osteoblasts is accurate and informative and, in my view, best represents an unbiased and broad view of the findings. Where is *Sp7* expression in the

transcriptomic analysis? This is a marker used throughout the paper in most figures and the authors use it in the final identification of OG2 as the tip-specific pre-osteoblast. Right now, it is only shown in supplementary figure but I would like to see it plotted in figure 4.

We now include UMAP visualization that shows broad *Sp7* expression in the osteogenic lineage in Fig. 5e.

Line 270. Text references fig 6c when it probably should be 5c. In the figure the legend with the color code lacks description (only numbers now) – please add what the numbers depict in the figure for increased readability.

This figure has now been moved to supplementary information (Supplementary Figure 10). The legend indicating that the numbers correspond to the counts of interactions has been added to the figure.

The section on “signaling interaction” is the weakest part of the manuscript. The authors have not studied the interaction between any of the mentioned molecules but speculate of their existence based on location of the cells that secrete them, a too superficial analysis based on numbers of potential interactions and prior knowledge. Given that one strong interaction could be enough for biological significance it is weird to make this statement based on number of potential interactions only. This argument would be somewhat strengthened if the authors at least would normalize for expression of possible interactions (i.e., perhaps MG and EC cells simply express more secreted factors in general?). Due to the superficial nature of this analysis, I believe that this part could be removed without lessening the impact of the study.

We agree that it is difficult to draw robust conclusions from this analysis and have therefore moved this figure to the supplemental information. While this is an imperfect analysis given the need for validation studies well beyond the present study, the finding that the ectocranial and meningeal layers closest to the suture have the highest number of predicted interactions with osteogenic cells supports our model for local paracrine interactions. We would also like to point out that identifying which populations express/secrete certain factors provides a really important framework for future studies, and for this reason have not removed the analysis entirely from the manuscript.

Figure 6: For figure 6a the authors should provide methods section explaining how 75% of cells can have 0.1 expression in light of that 10x provides UMIs. I guess this is because the authors are using data normalized in some way, but this should be clearly explained. Also seems like equally strong expression in midline as in coronal suture synostosis – main difference is percentage of cells expressing each gene. It is not clear beyond slightly better detection that difference in disease could be driven by these small differences. Given that 10x have quite low coverage is it possible that many negative

cells represent false negatives in the midline suture genes. Could this be statistically modelled?

As the reviewer points out, the data has been scaled. The Method section now includes details on how dotplots were generated: “ All included DotPlots reflect z-score scaled data generated from the DotPlot() function that enables visualization of lowly expressing clusters with highly expressing clusters on the same scale (<https://rdrr.io/bioc/scater/man/plotDots.html>). Dot size was determined by setting the parameters dot.min = 0 and dot.scale = 9. Expression scales were determined by setting the parameters minimal scale col.min = 0 and maximum scale col.max = 2.5.”

We have focused the analysis on the coronal suture and removed the comparative analysis to midline synostosis associated genes.

With regards to fig 6b I think the authors have made some weird choices given that many of the monogenetic causes are autosomal dominant. Why is an average calculated? Would it not be better to add signal on top of each other – expression of one mutated gene could be enough in an autosomal dominant disease.

Our goal was to assess in which cell clusters coronal synostosis genes were expressed, regardless of the mode of action of the mutations. While it is true that mutation of a single allele is often sufficient to cause craniosynostosis, we have simplified the approach by treating each gene similarly. This analysis tests if genes that cause coronal suture fusion are co-enriched within any of the captured cell types in our datasets. While it may be plausible to use the sum of signal to interrogate this expression, we prioritized an existing and well documented function within Seurat that utilizes averages to determine the coincidence of the synostosis modules. In both strategies, scores are impacted by amount of gene expression detected for each gene within a module by cell. Critically, this module enables score correction by assessing the average expression of random sets of genes to determine if the tested scores are greater than random chance (a positive module score). In new Supplementary Figure 12, we performed statistical analysis to ask whether coronal synostosis gene expression was non-randomly distributed in coronal, meningeal, and/or ectocranial clusters. This new analysis confirms that only the coronal gene set shows non-random enrichment in osteogenic clusters. Our approach is described in detail in the Methods section *Bioinformatics Analysis*.

Main text: To assess whether the coronal synostosis gene set had non-random distribution across suture cell types, we assessed the distribution of coronal gene set average expression, versus a randomized gene set, between grouped osteogenic, ectocranial, and meningeal clusters. Coronal synostosis genes were enriched in osteogenic clusters, and to a certain extent in meningeal clusters (Fig. 7b, Supplementary Fig. 12a). This analysis demonstrates a preferential enrichment of genes linked to coronal synostosis in osteogenic clusters.

For disease coupling the authors are also missing a golden opportunity to compare an independent data set from Holmes et al. Now they have hidden this away in supplementary figure 11 and only performed a very superficial analysis. I would suggest comparing not only midline vs coronal genes but also cell types from the two data sets.

We have now performed a much more detailed comparison of the two datasets, shown in new Fig. 6. Co-clustering shows that cell types are generally conserved between sutures, and this analysis has allowed us to assign comparable clusters with more rigor. We also looked at expression of the marker genes identified in the two studies within the composite dataset. Interestingly, OG1 specific genes (*Pthlh*, *Erg*, *Net1*) appear to be more highly expressed in the coronal versus frontal suture dataset.

Discussion

Line 298, the first paragraph of the discussion states that single cell sequencing bridges the current state of knowledge to provide answers on craniosynostosis. Although discovery sequencing is, in effect, and as shown by the authors, useful to know the parts list, it is insufficient to understand the pathogenesis at this point. I would focus on the presented results more as a step, mainly useful for researchers in the field, and less as the answer of what goes wrong in craniosynostosis, which this paper doesn't.

As discussed above, new analysis of a *Twist1/Tc12* mutant mouse model has allowed us to make new insights into the pathogenesis of this particular form of coronal synostosis – namely that OG1 progenitors are lost as early as E14.5. We now better discuss how this presents a paradigm for using the single-cell dataset to understand other forms of craniosynostosis.

Line 354: the claim that the expression of genes present in ligaments means that the ectocranial tissue acts like one is not justified and should be changed. Even if ligament or ligament like tissue was actually found that does not mean that it fulfills any tension bearing function. Ligaments are a common occurrence within the cranium and fulfill no such a function. These types of possibly or probable claim, in my view, weaken the paper. The authors provide an extraordinarily valuable resource for the research community, claims on the function on the different genes found should be supported on the available results.

New data using *Scx-GFP*, the gold standard for tendon and ligament fate, show that this ligament population persists above the coronal suture at least until 10 months after birth. We have also softened our claims that this ligament performs a tension-bearing function, now only suggesting it as a possibility in the Discussion (though we would be happy removing this if the reviewer feels strongly such speculation should be avoided).

Methods

The authors pooled 10 different mice, was each individual identified for analysis? Were there any outliers?

For the E15.5 analysis, embryos were pooled for a single 10X run. At E17.5, 3 batches of different embryos were analysed. The data from the 3 batches overlapped well, indicating there was no batch effect.

Reviewer #3 (Remarks to the Author):

This article is appropriate for publication in Nature Communication as it an insightful, well-presented and important contribution, however, minor comments remain.

We thank the reviewer for these very positive comments.

1. The images presented feature quite a lot of overexposed signal throughout. The saturated signal makes it tricky to appreciate the true extent of signal overlap and size of RNA loci. This is especially true for fig. 3. H, where a key point is that *Podnl1* is expressed within the bone edge but it is not clear whether these cells are also *Sp7* positive.

We recognize the reviewer's comment, and the intent of the exposure was to be sure to capture low *Sp7* expression at the growing tips to make stronger claims gene expression patterns within or around *Sp7*⁺ osteoblasts. To further ensure the reader can appreciate the gene expression relationships, in Supplementary Fig. 7h,i, we now present *Podnl1* in situs with and without *Sp7* expression. These additional images provide further evidence that this OG2 gene is generally excluded from the bone tips. Magnification of the merged *Sp7/Podnl1* images in this supplementary figure clearly shows at least some bone edge cells expressing both markers.

2. The figure legends could benefit from more detail, e.g. Fig.4. where a is listed simply as lineage analysis. For the uninitiated trainee, it might be helpful to point out that the grey line connects relationships between cell fates.

We have modified the Fig. 4 (now Fig. 5) legend (new text in italics) - Lineage analysis and **b** Pseudotime analysis of the osteoblast subset using Monocle 3. *The gray lines in a indicate the lineage relationship between cells in the trajectory. In b, cells further along the single cell trajectory are indicated by progressively lighter colors.* We have reviewed all the figure legends and added more detail where necessary.

3.Indeed, it may be helpful to further detail the parameters used in both Seurat clustering and the monocle based lineage analysis. While this reviewer understands there are recommendations within these packages, there is little procedural information on how the resolution of these clusters was chosen as an example. As these parameters can especially influence lineage associations (e.g. Fig 4.), more detail is advised.

We have expanded the bioinformatic methods section to include precise parameters for Seurat clustering and Monocle analysis as well as references for the tutorials used as guides to analyze our datasets. We have also provided the parameters for all visual depictions of single cell analysis.

4. As the numbers below the x axis in fig 4. c are too small to see and the legend includes that information, these could be deleted.

These numbers have been deleted as recommended, and the y-axis numbers have been enlarged.

Reviewers' Comments:

Reviewer #2:

Remarks to the Author:

I congratulate the authors on a much-improved study that I find highly valuable beyond simply a resource. The comparison between the two different types of sutures was indeed very interesting and the addition of both fate-mapping of Six2 cells adds another great aspect to the paper. Also, adding the details to the material and methods section improves the readability makes re-analysis possible. I recommend publication in Nature Communications.

Reviewer #3:

Remarks to the Author:

Understanding the cell biology of cranial sutures has remained elusive in both developmental biology and regenerative medicine due to the technical and practical challenges of a complex mesenchymal structure that forms relatively late in development and essential. Here, Farmer et al. take advantage of recent advancements in single-cell transcriptional assays to investigate the cellular complexity of developing sutures and in doing so identify one of the more convincing markers of suture progenitor identity throughout embryonic and early postnatal development, such as Six2. As one of many poorly understood mesenchymal stem cell niches in the vertebrate, parsing new markers for the many discrete subtypes the authors find is an important next step in this field that will become a resource for both developmental and regenerative studies. Excitingly, one suture subtype may represent a ligamentous tissue that was mapped through in situ analyses to lie above suture progenitors, not only in development but postnatal stages. Whether functional as a ligament or not, finding structural features such as a layer of ligament-like cells persisting through development advances our understanding not only of sutural complexity but perhaps also the physical environment that the suture generates to serve its function as a flexible joint. The authors also take advantage of their recently published and similar transcriptional analysis of frontal sutures to compare cell types between these structurally distinct sutures. Although they find overlapping cell types they find some differences in expression levels of characteristic markers exist and these cells have altered spatial organisation in frontal sutures compared to coronal. Having identified suture, bone, dermal and meningeal cell subtypes, the authors asked whether known drivers of premature suture fusion in humans were enriched in particular cell types, finding such markers to predominate in osteogenic cells and to a lesser extent the meninges. As these data suggest that particular cell types may indeed be altered in disease, the authors analyse a well-known model of coronal synostosis, *Twist*^{+/-}; *Tcf12*^{+/-} mutant mice, finding a specific reduction of *Erg* expressing cells which mark an osteogenic subtype, and the progenitor marker, Six2. Together, this submission presents an important resource for the diverse disciplines that use the suture as a model but also presents a framework to understand cellular organisation, differentiation, and signalling in this poorly understood and complex progenitor niche.

The authors have made significant improvements to an already exciting body of data that justifies release in the broadly appealing Nature Comms. While many single-cell transcriptomics studies can be accused of lacking mechanism, it is important to keep some perspective when considering what constitutes mechanism in tissue of this kind which has, until now, had too few markers to study its cell biology in 3D, let alone predict how different disease alleles may function to bring about premature suture fusion at the cell level. The clinical and developmental significance of unveiling the true complexity of this mesenchymal stem cell niche when many studies can only make tissue-level inferences should be commended, especially now as the authors integrate their previous work better with these data. Further, as human iPSC and patient genetic approaches in skull development and disease continue to push forward, spatial and transcriptomic mapping becomes a benchmark resource for tissue engineers. Although the authors may raise more questions than they definitively answer, this is not because of a lack of good science but rather is a reflection of the suture's complexity such that the authors now offer an important new platform through which 20 years of craniofacial and clinical genetics can be contextualised.

RESPONSE TO REVIEWER 1 COMMENTS:

1: Reviewer 1 was concerned that insufficient mechanistic data was presented to support the link between regulators of proliferation and synostosis that is also predicted throughout the literature.

The authors address this concern by analysing RNA seq data for indications of altered progenitor balance in mutant mice that represent a very common form of syndromic synostosis, Twist1/TCF12. These mice have been previously assayed for proliferation defects finding an increase in progenitor proliferation. This reviewer agrees with the conclusion of the authors that they present the most convincing quantitative indication that synostosis can be largely due to perturbation in progenitor number. However, I would like to add to their response by clarifying the importance of unbiased quantitative metrics of cell identity during morphogenesis, as were performed here. Quantifying proliferation/differentiation in sectioned samples is a necessary and essential path to understanding the development and the etiology of disease but it is the firm assertion of this reviewer based on the literature and this submission that there is more cellular complexity in the suture than simple designations our current markers allow. The beauty of this work is the unveiling of that complexity both transcriptionally and positionally that can allow so many to ask which progenitor is altered and how. Further, these data suggest that just like the disease, synostosis does not have to be driven through proliferation per se, but rather the differentiation of a progenitor which may undergo division as part of its program. Thus, highlighting how the balance of cell types is affected in disease can help us quantitatively understand the cellular behaviours as a whole and is the NEXT step in understanding complex organogenesis.

2. Reviewer 1 felt that temporal information was lacking, although two time points in development were analysed.

The authors have added developmental timepoints and further genotypes. While this reviewer felt that the authors had selected time points relevant to the literature and therefore within the scope of this paper, it is impressive that they were able to add another high-quality data set and I am grateful for the added richness that many will benefit from.

Minor comment:

Further methodological annotation has been added according to the comments of the reviewers.